# Glyoxal as an alternative fixative to formaldehyde in immunostaining and super-resolution microscopy

Katharina N Richter[1,2,†], Natalia H Revelo[1,†,‡] ID, Katharina J Seitz[1,3], Martin S Helm[1,3], Deblina Sarkar[4], Rebecca S Saleeb[5], Elisa D'Este[6], Jessica Eberle[7], Eva Wagner[8,9], Christian Vogl[10,11] ID, Diana F Lazaro[12,13], Frank Richter[3,14], Javier Coy-Vergara[15], Giovanna Coceano[16], Edward S Boyden[17], Rory R Duncan[5], Stefan W Hell[6], Marcel A Lauterbach[7], Stephan E Lehnart[8,9], Tobias Moser[10,11] ID, Tiago F Outeiro[12,13] ID, Peter Rehling[14,18], Blanche Schwappach[15], Ilaria Testa[16], Bolek Zapiec[19] & Silvio O Rizzoli[1,2,*] ID

## Abstract

**Paraformaldehyde (PFA) is the most commonly used fixative for immunostaining of cells, but has been associated with various problems, ranging from loss of antigenicity to changes in morphology during fixation. We show here that the small dialdehyde glyoxal can successfully replace PFA. Despite being less toxic than PFA, and, as most aldehydes, likely usable as a fixative, glyoxal has not yet been systematically tried in modern fluorescence microscopy. Here, we tested and optimized glyoxal fixation and surprisingly found it to be more efficient than PFA-based protocols. Glyoxal acted faster than PFA, cross-linked proteins more effectively, and improved the preservation of cellular morphology. We validated glyoxal fixation in multiple laboratories against different PFA-based protocols and confirmed that it enabled better immunostainings for a majority of the targets. Our data therefore support that glyoxal can be a valuable alternative to PFA for immunostaining.**

**Keywords** fixation; glyoxal; immunocytochemistry; PFA; super-resolution Microscopy

**Subject Categories** Methods & Resources
The EMBO Journal (2018) 37: 139–159

## Introduction

The 4% paraformaldehyde (PFA) solution has been a standard fixative for immunostaining and fluorescence microscopy, for several decades. Nevertheless, the literature contains numerous reports that PFA causes morphological changes, loss of epitopes, or mislocalization of target proteins and that it fixes the samples slowly and incompletely (see, e.g., Melan, 1994; Tanaka *et al*, 2010; Schnell *et al*, 2012). Many other fixatives have been introduced to alleviate these problems. Among them, glutaraldehyde is probably the most commonly used, since it fixes the samples faster and more completely than PFA (Smith & Reese, 1980). Mixtures of PFA and glutaraldehyde result in accurate fixation and reduce the lateral mobility of molecules (Tanaka *et al*, 2010), presumably by increasing the level of protein

1   Department of Neuro- and Sensory Physiology, University of Göttingen Medical Center, Göttingen, Germany
2   Cluster of Excellence Nanoscale Microscopy and Molecular Physiology of the Brain, Göttingen, Germany
3   International Max Planck Research School Molecular Biology, Göttingen, Germany
4   MIT Media Lab
5   Edinburgh Super-Resolution Imaging Consortium, Institute of Biological Chemistry, Biophysics, and Bioengineering, Heriot-Watt University, Edinburgh, UK
6   Department of NanoBiophotonics, Max-Planck-Institute for Biophysical Chemistry, Göttingen, Germany
7   Department of Neural Systems, Max-Planck-Institute for Brain Research, Frankfurt am Main, Germany
8   Heart Research Center Göttingen, Department of Cardiology & Pulmonology, University Medical Center Göttingen, Göttingen, Germany
9   German Center for Cardiovascular Research (DZHK) Site Göttingen
10  Institute for Auditory Neuroscience and InnerEarLab, University Medical Center Göttingen, Göttingen, Germany
11  Max-Planck-Institute for Experimental Medicine, Auditory Neuroscience Group, Göttingen, Germany
12  Department of Experimental Neurodegeneration, Center for Nanoscale Microscopy and Molecular Physiology of the Brain, Center for Biostructural Imaging of Neurodegeneration, University Medical Center Göttingen, Göttingen, Germany
13  Max-Planck-Institute for Experimental Medicine, Göttingen, Germany
14  Department of Cellular Biochemistry, University Medical Center Göttingen, Göttingen, Germany
15  Department of Molecular Biology, University Medical Center Göttingen, Göttingen, Germany
16  Department of Applied Physics and Science for Life Laboratory, KTH Royal Institute of Technology, Stockholm, Sweden
17  Departments of Brain and Cognitive Science and Biological Engineering, MIT Media Lab and McGovern Institute, Cambridge, MA, USA
18  Max-Planck-Institute for Biophysical Chemistry, Göttingen, Germany
19  Max Planck Research Unit for Neurogenetics, Frankfurt am Main, Germany
    *Corresponding author. Tel: +49 551 395911; E-mail: srizzol@gwdg.de
    †These authors contributed equally to this work
    ‡Present address: Department of Tumor Immunology, Radboud Institute for Molecular Life Sciences, Radboud University Medical Center, Nijmegen, the Netherlands

cross-linking. However, this fixative mixture also reduces the efficiency of immunostainings, by blocking the antibody access to epitopes, or by causing particular epitopes to unfold (Farr & Nakane, 1981). Alcohol-based fixation, such as treatments with ice-cold methanol (Tanaka *et al*, 2010), results in stable fixation for a subpopulation of cellular structures (such as microtubules), but leads to poor morphology preservation and to a loss of membranes and cytosolic proteins. Overall, the improvements in fixation induced by glutaraldehyde or methanol do not compensate for their shortcomings, thus in most cases leaving PFA as the current fixative of choice.

A superior alternative to PFA is needed, especially since artifacts that were negligible in conventional microscopy are now rendered visible by the recent progress in super-resolution microscopy (nanoscopy; Eggeling *et al*, 2015). To find a fixative that maintains high-quality immunostainings while alleviating PFA problems, we have tested several compounds. We searched for commercially available molecules, which could be readily used by the imaging community. These included different combinations of PFA and glutaraldehyde, picric acid (Hopwood, 1985), and di-imido-esters (Woodruff & Rasmussen, 1979), which, however, were not better than PFA in immunostaining experiments. We have also investigated different aldehydes. We avoided highly toxic compounds such as acrolein, which would not be easy to use in biology laboratories, and we also avoided large aldehydes (more than 4–5 carbon atoms), whose fixative properties are expected to mimic those of glutaraldehyde. The small dialdehyde glyoxal fits these two criteria, since it has a low toxicity (as already noted in the 1940s, Wicks & Suntzeff, 1943) and contains only two carbon atoms. Glyoxal is used, at low concentrations, in glycation and metabolism studies (Boucher *et al*, 2015), which ensures that it is commercially available. It can be used as a fixative and has even been once described, in 1963, to provide better morphology preservation to formaldehyde (Sabatini *et al*, 1963). It is almost unknown in fluorescence experiments. We were able to find one publication, from 1975 (Swaab *et al*, 1975), in which glyoxal was used in immunofluorescence on brain samples, albeit followed by sample freezing, and by procedures that are not compatible with modern, high-quality microscopy. We could also find a few publications on histological stains using glyoxal (e.g., Umlas & Tulecke, 2004; Paavilainen *et al*, 2010), which further encouraged us to test this compound.

We tested glyoxal thoroughly, in preparations ranging from cell-free cytosol to tissues, and by methods spanning from SDS–PAGE to electron microscopy and super-resolution fluorescence microscopy. We found that glyoxal penetrated cells far more rapidly than PFA and cross-linked proteins and nucleic acids more strongly, leading to a more accurate preservation of cellular morphology. Despite the stronger fixation, glyoxal did not cause a reduction of antibody binding to the samples. On the contrary, the resulting images were typically brighter than those obtained after PFA fixation. The initial optimization work was performed in one laboratory (Rizzoli, University Medical Center Göttingen, Germany), and the results were independently tested in 11 additional laboratories/teams: Boyden (MIT Media Lab and McGovern Institute, Massachusetts, USA), Duncan (Heriot-Watt University, Edinburgh, UK), Hell (Max-Planck-Institute for Biophysical Chemistry, Göttingen, Germany), Lauterbach (Max-Planck-Institute for Brain Research, Frankfurt am Main, Germany),

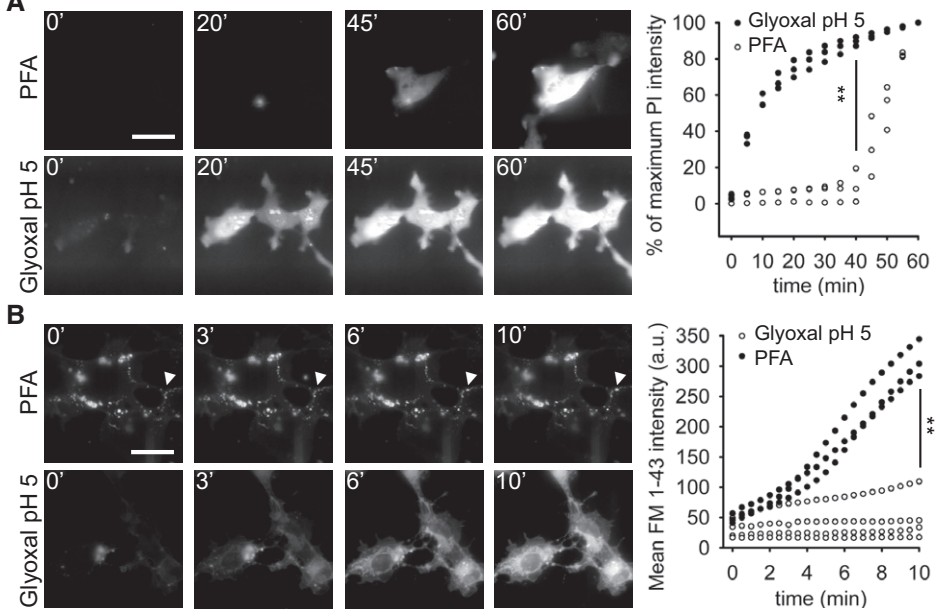

**Figure 1.  Comparison of cell penetration by PFA and glyoxal.**

A   Speed of propidium iodide (PI) penetration into fibroblasts during 60 min of fixation with either 4% PFA or 3% glyoxal. *N* = 3 independent experiments. Glyoxal fixation enables PI to penetrate far more rapidly into the cells.

B   Speed of FM 1-43 penetration in similar experiments. The arrowhead points to one example of ongoing endocytosis during PFA fixation. *N* = 3–4 independent experiments. The general pattern of FM 1-43 entry was similar to that of propidium iodide. Only the first 10 min are shown, to enable an optimal observation of the kinetics of the first stages of FM 1-43 entry. The results parallel those obtained with PI: faster penetration during glyoxal fixation.

Data information: Scale bar = 40 μm; **P < 0.01 (two-sided Student's *t*-test).

Lehnart (University Medical Center Göttingen, Germany), Moser (University Medical Center Göttingen, Germany), Outeiro (University Medical Center Göttingen, Germany), Rehling (University Medical Center Göttingen, Germany), Schwappach (University Medical Center Göttingen, Germany), Testa (KTH Royal Institute of Technology, Stockholm, Sweden), and Zapiec (Max Planck Research Unit for Neurogenetics, Frankfurt am Main, Germany). We conclude that the immunostainings performed after glyoxal fixation were superior for the majority of the samples and targets, with only a minority (~10%) of the targets being less well preserved and/or revealed.

## Results

### Glyoxal preserves the cellular morphology more accurately than PFA and fixes proteins and RNAs more strongly

To determine the optimal conditions of glyoxal fixation, we tested its action at different pH values (Appendix Table S1). We found that glyoxal requires an acidic pH, roughly between 4 and 5, despite one previous study that suggests that it may also fix samples at a neutral pH (Sabatini *et al*, 1963). In addition, we found that the morphology of the samples was much improved upon addition of a low-to-medium concentration of alcohol (ethanol, 10–20%), which may act as an accelerator in the fixation reactions. Removing the ethanol, or adjusting the pH above or below the 4–5 range, resulted in poor sample morphology (Appendix Table S1). pH values of 4 or 5 provided similar results for most of our experiments (results obtained

at pH 5 are shown in all figures, unless noted otherwise) and provided better morphology preservation for cultured neurons than PFA. We tested PFA at various pH values (4, 5, and 7), with or without ethanol, at room temperature or at 37°C (Appendix Table S1), without finding a condition where the morphology of the PFA-fixed samples consistently bettered that of glyoxal-fixed samples.

We then proceeded to compare PFA and glyoxal fixation quantitatively. We first tested the speed with which these fixative solutions penetrate the cell membrane, by monitoring the fluorescence of propidium iodide, a fluorogenic probe that binds nucleic acids, and cannot enter living cells (Davey & Kell, 1996). Paraformaldehyde fixation allowed propidium iodide entry into cultured cells only after ~40 min, while glyoxal was substantially faster (Fig 1A). The same was observed using the membrane-impermeant styryl dye FM 1-43 (Betz *et al*, 1992): Glyoxal

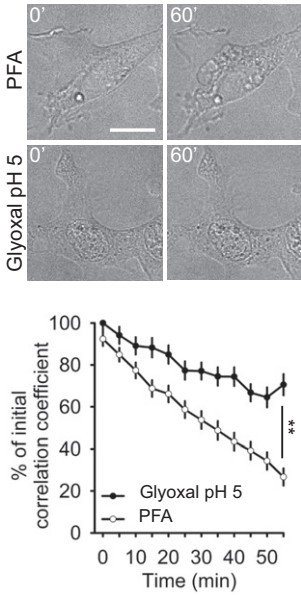

**Figure 2.  A comparison of morphological changes taking place during fixation with PFA or glyoxal.**

The changes were visualized by DIC images taken at 5-min intervals during fixation. The graph shows the correlation of each image to the first frame. *N* = 50 (PFA) and 54 (glyoxal) cellular regions analyzed, from three independent experiments (mean ± SEM). The higher correlation value indicates that glyoxal preserves the initial cell morphology with higher accuracy than PFA. Scale bar = 20 μm; **P < 0.01 (two-sided Student's *t*-test).

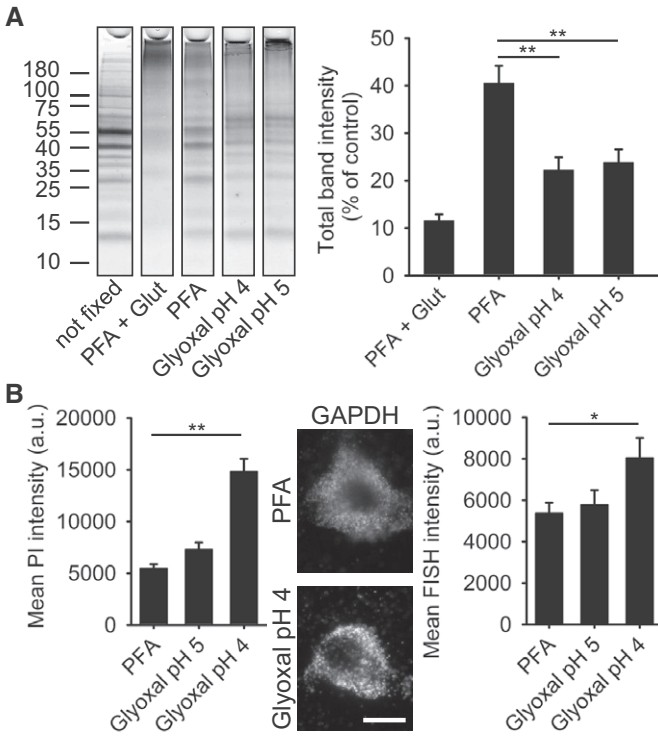

**Figure 3.  Comparison of protein and RNA fixation by PFA and glyoxal.**

A   SDS–PAGE gel showing rat brain cytoplasm incubated for 60 min with different fixatives. The graph shows the summed intensity of the bands in each lane. Fixed proteins either no longer run into the gel or form only smears. To compare the efficiency of fixation, the bands that survive fixation were summed and were expressed as % of an unfixed control. The intensity of PFA-fixed samples was significantly higher than that of glyoxal-fixed samples (*N* = 5 independent experiments; one-way ANOVA with *post hoc* Tukey test). Glut = 0.2% glutaraldehyde.

B   Staining of nucleic acids after fixation. The propidium iodide signal in fibroblasts was significantly higher for samples fixed with glyoxal pH 4 (*N* = 6–8). To test whether the fixed nucleic acids were still available for specific detection, we performed FISH for GAPDH in cultured neurons, using a standard protocol provided by the company Affymetrix. The fluorescence signal of the samples fixed with glyoxal (pH 4) was significantly higher than for PFA-fixed samples (*N* = 5–6; two-sided Student's *t*-test).

Data information: The graphs show mean values, and error bars represent standard error of the mean. Scale bar = 10 μm; *P < 0.05, **P < 0.01.

fixation enabled significant FM 1-43 penetration within 1–2 min (Fig 1B). The difference in membrane penetration is probably due to the ethanol present in the glyoxal fixative, since the addition of ethanol to the PFA solution enhances its penetration into cells in a similar fashion (Appendix Fig S1), albeit it did not improve immunostainings with PFA (Appendix Fig S1; we would like to point out that low pH values, 4 and 5, also failed to improve PFA immunostainings, as shown in the same Appendix figure). In the same experiments, FM 1-43 addition enabled us to visualize endocytotic events that took place during PFA fixation. Such events could be observed in every fixed cell (Fig 1B) and indicated that the cells were still active during PFA fixation, from the point of view of membrane trafficking. No such events could be detected during glyoxal fixation.

The hypothesis that cells were still partially active during PFA fixation, and less so in glyoxal fixation, was also confirmed by other experiments. First, we tested whether transferrin, which is readily endocytosed by a clathrin-mediated pathway, through the involvement of the transferrin receptor, is internalized during fixation. We applied fluorescently conjugated transferrin onto cells during fixation with glyoxal or PFA (Appendix Fig S2). We found that it was mainly fixed onto the plasma membrane by glyoxal, but that it was present both in the cells and on the membrane during PFA fixation (Appendix Fig S2). Second, we tested whether the acidic lumen of the lysosome was maintained after fixation, by applying the probe LysoTracker (Appendix Fig S3). Substantial LysoTracker labeling was observed after PFA fixation, but not after glyoxal fixation. Both of these experiments, therefore, indicate that glyoxal fixation stops cellular functions more efficiently than PFA.

The higher speed of membrane penetration seen with glyoxal was coupled to a better preservation of the general cell morphology, as observed by imaging cells during fixation (Fig 2). Paraformaldehyde fixation was associated with the formation of membrane blebs and vacuoles, with organelle movement, and with a general change in the cell morphology (Fig 2). Glyoxal fixation appeared to modify the cell morphology far less. This impression was confirmed by calculating the correlation coefficient between the initial cell images and images acquired at 5-min intervals during fixation (Fig 2). To obtain a similar view at the level of single organelles, we imaged the movement of endosomes labeled with fluorescently conjugated transferrin or cholera toxin. As for the general cell morphology,

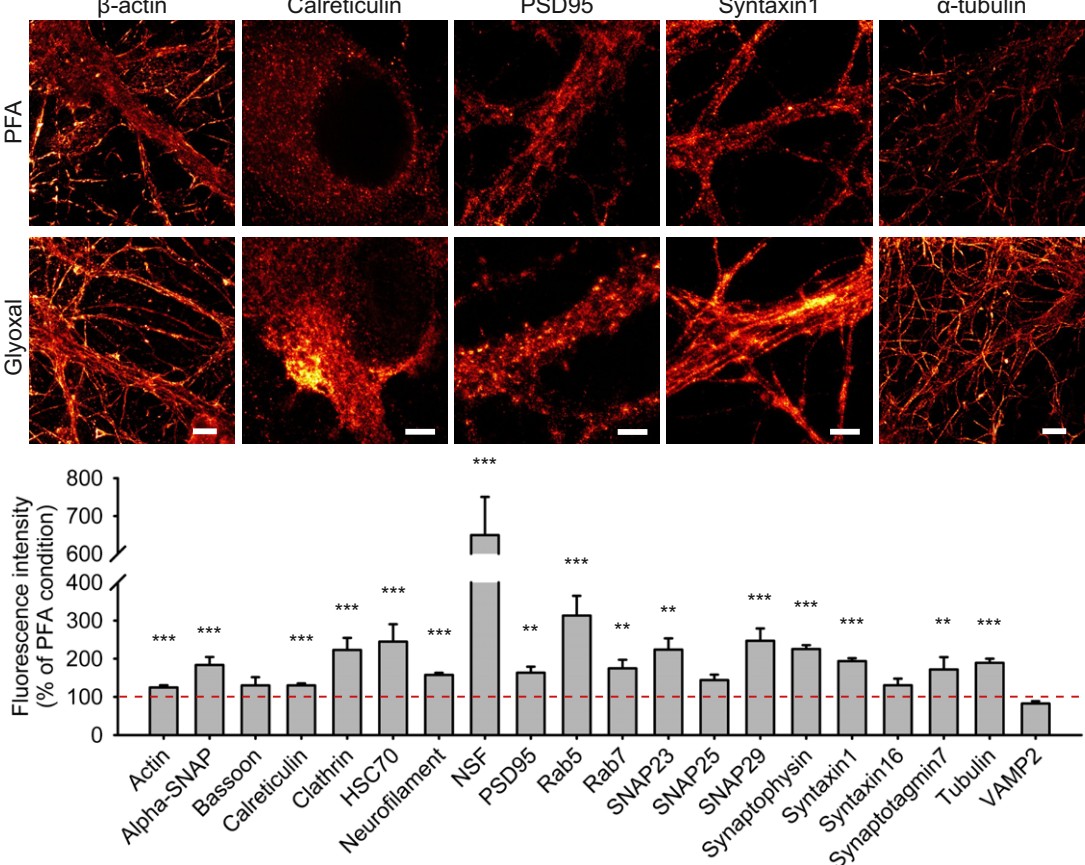

**Figure 4.  STED imaging of primary hippocampal neurons fixed with either PFA or glyoxal.**
Strong differences in labeling patterns can be observed. The images are brighter and less spotty for the glyoxal-fixed samples. Structures such as filaments or organelles are more easily detected. Quantification of the fluorescence signal (fold over background) shows that 16 out of 20 stainings are significantly brighter in glyoxal-fixed samples compared to the PFA-fixed samples. $N = 35$–$132$ objects (mean $\pm$ SEM). Scale bar = 6 μm for β-actin and α-tubulin and 3 μm for the other proteins. $**P < 0.01$, $***P < 0.001$ (two-sided Student's $t$-test for PSD95, Wilcoxon rank-sum test for all other proteins).

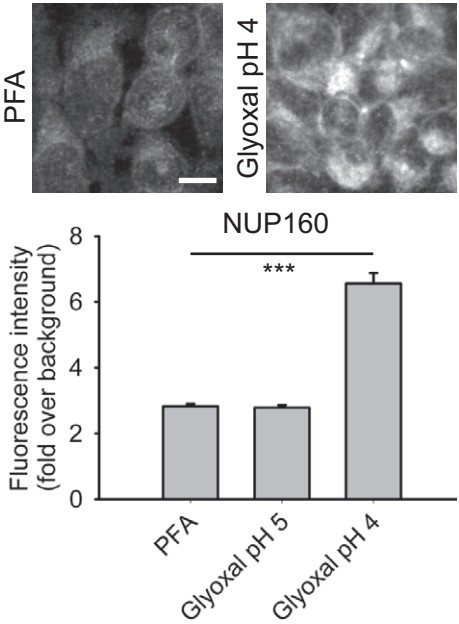

**Figure 5. Comparison of immunostaining NUP160 after fixation with either PFA or glyoxal.**

HeLa cells were stained for the nucleoporin complex protein NUP160 after fixation with either PFA, glyoxal pH 4 or glyoxal pH 5. Fluorescence intensities (fold over background) were compared and are shown in the graph. The quantification of fluorescence signals shows that glyoxal pH 4 fixation allows for significantly brighter stainings. $N = 73–156$ cells per condition analyzed (mean ± SEM). Scale bar = 10 μm. ***$P < 0.001$ (Wilcoxon rank-sum test).

glyoxal reduced the organelle movement more than PFA (Appendix Fig S4).

We also monitored the morphology of mitochondria, which are known to become fragile during fixation. We visualized mitochondria in living cells, by tagging them with a GFP-linked reporter (TOMM70, Appendix Fig S5), and imaged them again after fixation. Glyoxal preserved mitochondria at least as well as PFA. Moreover, ethanol addition to the PFA solutions worsened the preservation of mitochondria morphology, which suggests that ethanol does not improve PFA fixation, although it enhances its membrane penetration (Appendix Fig S5). To test this issue further, we analyzed the correlation between the pre- and post-fixation images for fluorescent protein chimeras of a mitochondria reporter (TOMM70), a Golgi apparatus reporter (GalNacT2), a plasma membrane reporter (SNAP25), a cytoskeleton reporter (tubulin), and a vesicular reporter (synaptophysin). The correlations were similar among the two fixatives for TOMM70, GalNacT2, tubulin, and SNAP25. However, the pre- and post-fixation correlations in glyoxal fixed samples were higher for synaptophysin (Appendix Fig S6), which marks the most mobile elements we investigated in this experiment (vesicles).

We then tested the protein cross-linking capacity of the different fixatives, by monitoring the proportion of the proteins that remained unfixed. We incubated brain cytosol samples with different fixatives for 60 min and followed this by running the samples on polyacrylamide gels (Fig 3A, Appendix Fig S7). Paraformaldehyde, with or without ethanol addition, left ~40% of the proteins unaffected (unfixed). Glyoxal (both pH 4 and 5) reduced this unfixed pool to ~20%. Shorter fixation times reduced

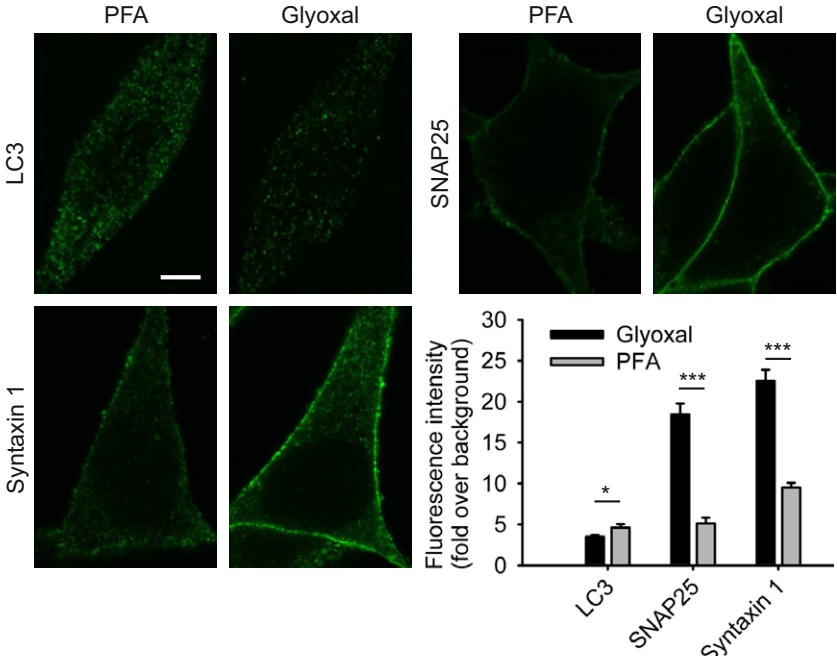

**Figure 6. Comparison of immunostained AtT20 cells after fixation with either PFA or glyoxal.**

AtT20 cells stained for the SNARE proteins syntaxin 1 and SNAP25 and the autophagy marker LC3B were compared with regard to the fluorescence intensity (fold over background) of the stainings. Quantification of the intensity shows that glyoxal fixation allows for significantly brighter stainings of the membrane SNARE proteins. LC3B staining is brighter in PFA-fixed cells. $N = 9–20$ cells per condition (mean ± SEM). Scale bar = 5 μm. *$P < 0.05$, ***$P < 0.001$ (two-sided Student's *t*-test).

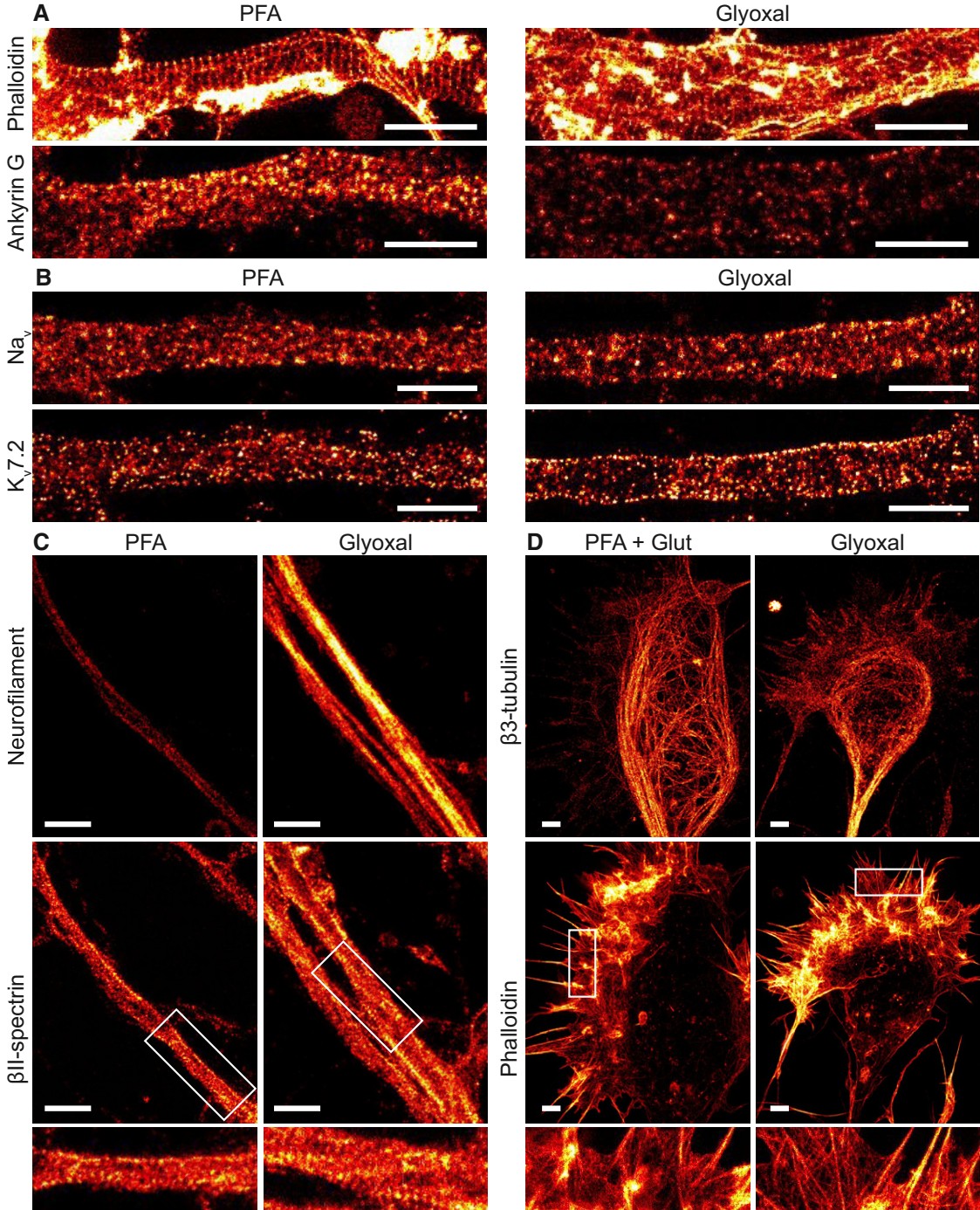

**Figure 7.  Comparison of immunostained primary hippocampal neurons in STED resolution.**

A   Primary hippocampal neurons were stained for actin and ankyrin G. A comparison between PFA- and glyoxal-fixed samples shows that actin staining with phalloidin works at least as well in both, showing the prominent actin rings. Ankyrin G staining is brighter in PFA-fixed cells.

B   Primary hippocampal neurons were stained for pan-$Na_v$ and $K_v7.2$. Both stainings seem to work at least as well for glyoxal-fixed neurons as for PFA-fixed neurons. Staining of K-channels shows a slightly more regular pattern in glyoxal-fixed neurons.

C   Primary hippocampal neurons were stained for neurofilament L and beta II spectrin. While the spectrin staining seems to be equally well in both fixation conditions, neurofilament staining is brighter in glyoxal-fixed cells.

D   Growth cones of hippocampal neurons were stained for actin and βIII-tubulin after either glyoxal or PFA + glutaraldehyde fixation. The latter is a standard fixation used for the co-labeling of tubulin and actin and is a stronger fixation than normal PFA fixation, which is incompatible with many organelle immunostainings (unlike glyoxal fixation). The filopodia and lamellipodia of the growth cones seem to be well stained for the samples fixed with glyoxal, whereas the samples fixed with PFA and glutaraldehyde seem to have lost some of the finer actin structures. Tubulin seems to be a bit better stained in samples fixed with PFA and glutaraldehyde.

Data information: Scale bars = 1 μm.

Sepia Fin FMRFamide

PFA                 Glyoxal

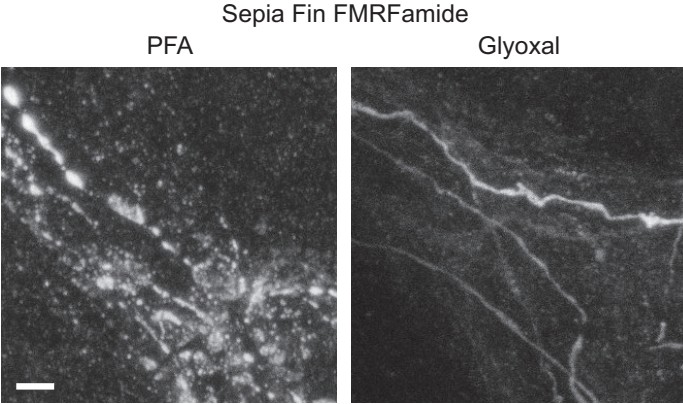

**Figure 8.   Comparison of immunostained sepia fin after fixation with either PFA or glyoxal.**

Sepia fin was fixed with the respective fixative and stained for the neuropeptide FMRFamide. A clear change in morphology can be observed between samples fixed with PFA and samples fixed with glyoxal. The former appear broken and swollen, while the glyoxal-fixed ones appear complete. The effect is presumably due to the different speed of penetration into tissue and/or fixation. Scale bar = 5 μm.

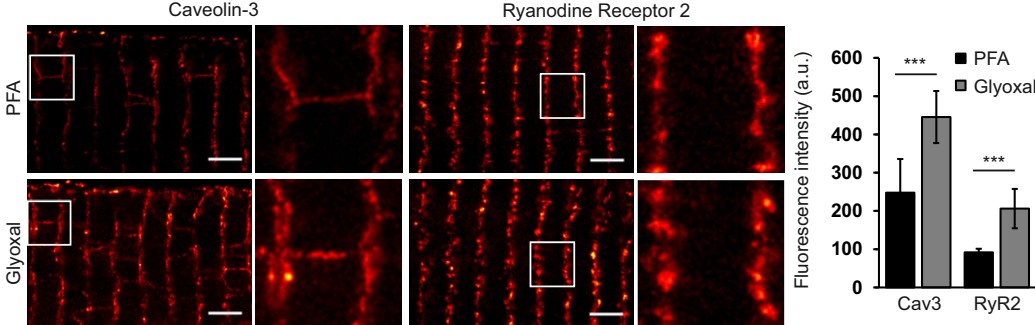

**Figure 9.   Comparison of immunostained ventricular myocytes fixed 10 min with either PFA or glyoxal.**

Freshly isolated murine ventricular myocytes were either fixed with 4% PFA or 3% glyoxal and immunostained for caveolin-3 or ryanodine receptor type 2. Quantification of the fluorescence intensity of the stainings shows significantly brighter stainings for glyoxal-fixed myocytes. The graph shows mean values, and error bars represent standard deviations. $N$ = 10 (RyR2) and 12 (Cav3) myocytes per condition. Scale bar = 2 μm. ***$P$ < 0.001 (Wilcoxon rank-sum test).

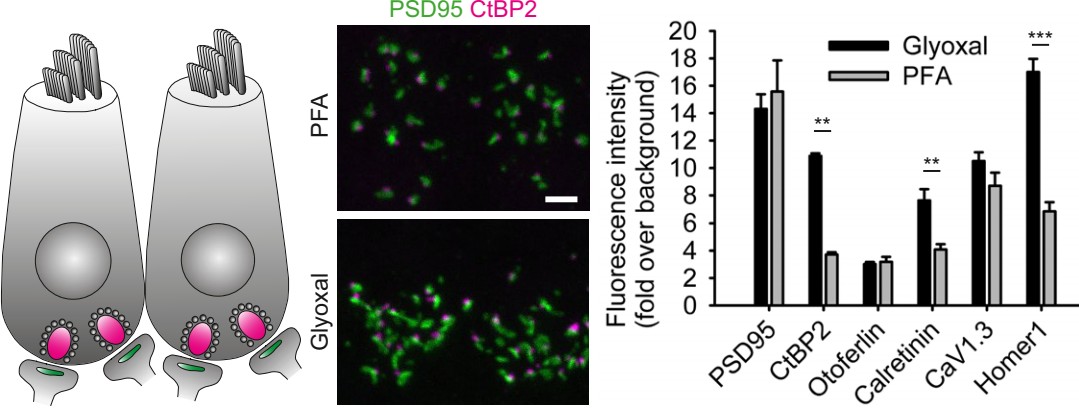

**Figure 10.   Comparison of immunostained mouse inner hair cells after fixation with either PFA or glyoxal.**

Acutely dissected organs of Corti were fixed in the respective fixative and immunostained for inner hair cell proteins and synaptic proteins. The quantification of fluorescence intensity for each staining shows a significant increase in signal-to-noise ratio for three target proteins (CtBP2, calretinin, and Homer 1) fixed with glyoxal. None of the stained proteins shows a significant decrease in fluorescence after glyoxal fixation. Representative images show maximum intensity projections from z-stacks of inner hair cell ribbon synapses. $N$ = 5 independent stainings from two animals per condition (PSD95, CtBP2, otoferlin, calretinin) and 10–15 images per condition (CaV1.3 and Homer 1) (mean ± SEM). Scale bar = 2 μm. **$P$ < 0.01, ***$P$ < 0.001 (two-sided Student's $t$-test for CtBP2, otoferlin, calretinin and Homer1, Wilcoxon rank-sum test for PSD95 and CaV1.3).

the amount of fixed proteins for all fixation conditions (Appendix Fig S7). Glyoxal, both at pH 4 or at pH 5, fixed more proteins than PFA, PFA and ethanol or PFA at low pH, at all time points (Appendix Fig S7).

The stronger fixation by glyoxal also applied for RNA molecules, albeit only at pH 4, as observed by staining cells with propidium iodide after fixation (Fig 3B). To test whether the glyoxal-fixed RNA molecules could still be detected by specific labeling, we performed fluorescence *in situ* hybridization (FISH) for a target that is often used as a standard in such experiments, glyceraldehyde 3-phosphate dehydrogenase (GAPDH). As for the propidium iodide staining, the GAPDH signal intensity was significantly raised by glyoxal at pH 4 (Fig 3B).

To test whether similar effects apply also to lipids, we immunostained cultured cells for phosphatidylinositol-(4,5)-P2 (PIP$_2$). The intensity of the immunostaining was substantially higher after glyoxal fixation (Appendix Fig S8).

The stronger fixation induced by glyoxal could be a concern for experiments relying on enzymatic tags, such as the SNAP-tag (Xue *et al*, 2015). Strong fixation may damage the enzyme, which would result in limited labeling. To test this, we expressed proteins coupled to the SNAP-tag in cultured cells, fixed them with PFA or glyoxal, and then incubated them with a fluorophore that is bound by the SNAP-tag, which couples to it covalently. The intensity of

glyoxal-fixed samples was significantly higher than that of PFA-fixed samples (Appendix Fig S9).

### Glyoxal provides higher-quality STED images in immunostaining than PFA

Having verified that glyoxal is a faster and more effective fixative than PFA, we proceeded to investigate its efficiency in immunostaining. We expressed fluorescent protein chimeras of reporters for mitochondria, the Golgi apparatus, the plasma membrane, the cytoskeleton, and vesicles, fixed the cells with PFA or glyoxal, and immunostained them. The immunostaining intensity of all of these structures, defined by the fluorescent protein signals, was significantly higher after glyoxal fixation (Appendix Fig S10). To also test this without the expression of fluorescently tagged proteins, we immunostained cells that had been incubated with fluorescently labeled transferrin (Appendix Fig S11). The transferrin is in this case present in endosomes, which should co-localize with endosomal markers such as EEA1. The co-localization was substantially higher after glyoxal fixation (Appendix Fig S11).

We then focused on cultured neurons, which have been a standard preparation for nanoscopy (Willig *et al*, 2006; Xu *et al*, 2013), and found that the resulting images were often brighter (Fig 4). We

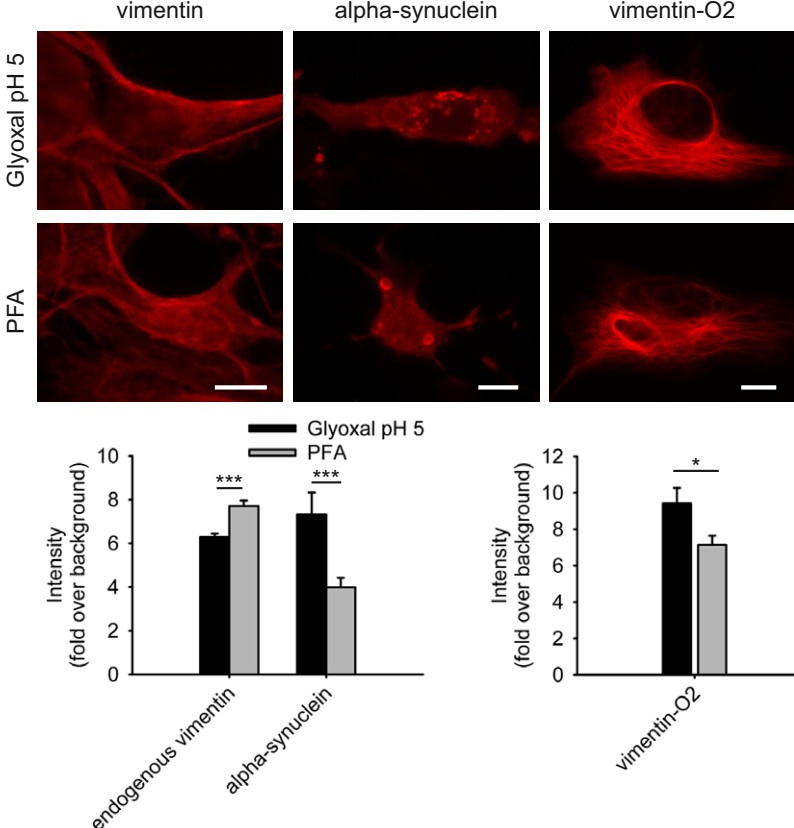

**Figure 11.  Comparison of stained vimentin and α-synuclein in human neuroglioma cells after fixation with either PFA or glyoxal.**
Cells were fixed with PFA or glyoxal for 10 min and were stained for endogenous vimentin, or for expressed α-synuclein. A quantification of the staining intensities indicates that glyoxal fixation allows for significantly brighter stainings for alpha-synuclein, but that PFA was superior for endogenous vimentin (leftmost graph). The fluorescence intensity of vimentin expressed with an mOrange2 tag was also analyzed after fixation with PFA or with glyoxal; the latter allowed more mOrange2 fluorescence to be detected (rightmost graph). N = 29–81 cell regions per condition (mean ± SEM). *P < 0.05, ***P < 0.001 (Wilcoxon rank-sum test). Scale bar: 10 μm.

analyzed the intensity of the STED images, in terms of signal over background (Fig 4), and determined that glyoxal indeed provided a higher signal for the large majority of the neuronal targets we investigated.

When investigated by STED microscopy, the many images of PFA-fixed cells appeared dominated by isolated, uniformly distributed spots, which presumably represent antibody clusters (Lang & Rizzoli, 2010; Opazo *et al*, 2012). The immunostaining signals appeared to be grouped in less uniform, more organelle-like structures after glyoxal fixation. To quantify this impression, an experienced user counted the number of organelle-like structures per $\mu m^2$, in a blind fashion, for 20 targets immunostained in neurons. This provided a quantitative (albeit user-driven) measurement of the accuracy of the stainings (Appendix Fig S12). This analysis suggested that the immunostainings performed after glyoxal fixation more readily allow the identification of organelles, for the majority of the targets. This impression was confirmed by several additional analyses (see Appendix Figs S13 and S14).

A possible cause for the appearance of isolated, uniformly distributed spots in the PFA-fixed samples is the loss of some of the unfixed soluble molecules after PFA fixation, through diffusion into the extracellular space. As indicated in Fig 3, 40% of the proteins remained unfixed and could therefore diffuse from the samples. To further test this hypothesis, we analyzed hippocampal neurons in electron microscopy, after fixation with PFA or glyoxal (Appendix Fig S15). The cytosol appeared clearer (less electron-dense) in the PFA-fixed samples. In contrast, the glyoxal-fixed samples had a more electron-dense cytosol, which rendered them similar, at least superficially, to samples prepared by high-pressure freezing (Appendix Fig S15).

We concluded, so far, that glyoxal appeared to be more efficient than PFA in several ways, such as speed and morphology preservation, which rendered it a better fixative for immunostaining and nanoscopy. Albeit we focused so far mostly on cell cultures, we also tested glyoxal in tissue preparations, where it enabled us to perform accurate immunostainings, in both *Drosophila* and mouse (Appendix Figs S16–S18). We did not observe any difficulties in the antibody penetration in such tissues, in contrast to fixation by, for example, glutaraldehyde (as discussed in the Introduction).

## Glyoxal provides higher-quality images in immunostaining for many different laboratories

The glyoxal fixation procedure established above was then tested in 11 different laboratories, in four countries (Germany, Sweden, United Kingdom, and United States). We present the results in alphabetical order.

The Boyden laboratory (MIT Media Lab and McGovern Institute, Departments of Brain and Cognitive Science and Biological Engineering, Cambridge, Massachusetts, USA) tested nucleoporin 160 in conventional immunostainings of cell cultures and found that fixation with glyoxal at pH 4 resulted in brighter images than those obtained with PFA fixation. The samples exhibited similar morphology (Fig 5).

The Duncan laboratory (Edinburgh Super-Resolution Imaging Consortium, Institute of Biological Chemistry, Biophysics, and Bioengineering, Heriot-Watt University, Edinburgh, UK.) also used conventional immunostainings of cultured cells (AtT20 cells) and analyzed the SNARE proteins syntaxin 1 and SNAP25 and the autophagy marker LC3B. The immunostainings of the two SNARE proteins were brighter after glyoxal fixation (Fig 6), whereas LC3B

staining was brighter after PFA fixation. The morphology of the cells appeared similar for the two fixatives.

The Hell laboratory (Department of NanoBiophotonics, Max-Planck-Institute for Biophysical Chemistry, Göttingen, Germany) used 3D STED microscopy to analyze the organization of several cytoskeletal proteins and of two membrane channels in axons and in growth cones of rat hippocampal cultured neurons (Fig 7). Actin was labeled using phalloidin, while all other proteins were labeled by immunostaining with previously published antibodies. Phalloidin stainings were similar after PFA or glyoxal fixation in axons (Fig 7A), but glyoxal revealed fine structures in growth cones that were not visible even after strong fixation with PFA and glutaraldehyde (Fig 7D). Neurofilaments were brighter after glyoxal fixation (Fig 7C), while another cytoskeletal element, ankyrin G, was brighter for PFA fixations (Fig 7A). The cytoskeletal protein βII-spectrin was

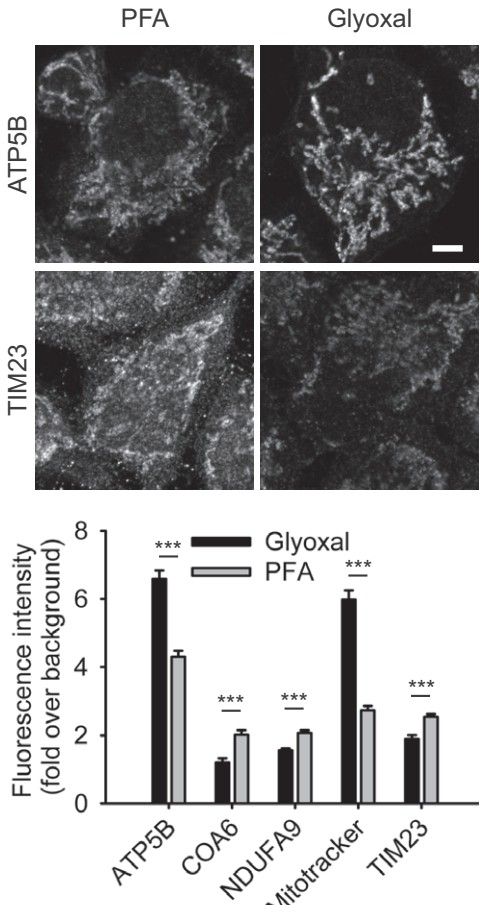

**Figure 12. Comparison of immunostainings for mitochondrial proteins after fixation with either PFA or glyoxal.**

Cells were stained with MitoTracker Orange prior to fixation with the respective fixative and immunostained for the mitochondrial proteins ATP5B, COA6, NDUFA9, and TIM23. Quantification of the staining intensity shows a significant increase of fluorescence (signal over background) for two markers (ATP5B and MitoTracker) after fixation with glyoxal, whereas for the remaining three proteins, immunostainings seem to be more efficient after fixation with PFA, albeit the differences are small. $N$ = 18–128 cells per condition (mean ± SEM). Scale bar = 5 μm. ***$P$ < 0.001 (two-sided Student's *t*-test for ATP5B and NDUFA9, Wilcoxon rank-sum test for all other proteins).

equally well stained in PFA or glyoxal fixation (Fig 7C); the same was observed for voltage-gated sodium channels (Fig 7B). The stainings for voltage-gated potassium channels were somewhat more regular for glyoxal fixation (Fig 7B). Finally, the staining of tubulin was marginally less bright for glyoxal fixation when compared with a protocol optimized for tubulin stainings (fixation with PFA and glutaraldehyde), and dynamic microtubules in the growth cones were better preserved with that fixation (Fig 7D).

The Lauterbach team (Max-Planck-Institute for Brain Research, Frankfurt am Main, Germany) focused on sepia fin immunostainings, testing the organization of FMRFamide in axons. Paraformaldehyde fixation resulted in poorer morphology, with fragmented axons, while glyoxal fixation revealed axons that appeared physiologically normal (Fig 8).

The Lehnart laboratory (Heart Research Center, Department of Cardiology & Pulmonology, University Medical Center Göttingen,

Germany) analyzed the trafficking and signal domain scaffolding protein caveolin-3 and the ryanodine receptor (of the sarcoendoplasmic reticulum) in ventricular myocytes isolated from mouse hearts, using STED microscopy. The immunostaining intensity of both of these proteins was significantly higher for glyoxal fixation (Fig 9). Of note, the morphology remained similar, as determined by 3D STED microscopy.

The Moser laboratory (Institute for Auditory Neuroscience and InnerEarLab, University Medical Center Göttingen; Auditory Neuroscience Group, Max-Planck-Institute for Experimental Medicine Göttingen, Germany) analyzed several proteins in the synapses formed between mouse cochlear inner hair cells and afferent spiral ganglion neurons (Fig 10). No substantial differences were found for the active zone protein PSD95, for the trafficking protein otoferlin, and for the presynaptic voltage-gated calcium channel (Ca$_V$1.3). The signal intensities were substantially higher after glyoxal fixation

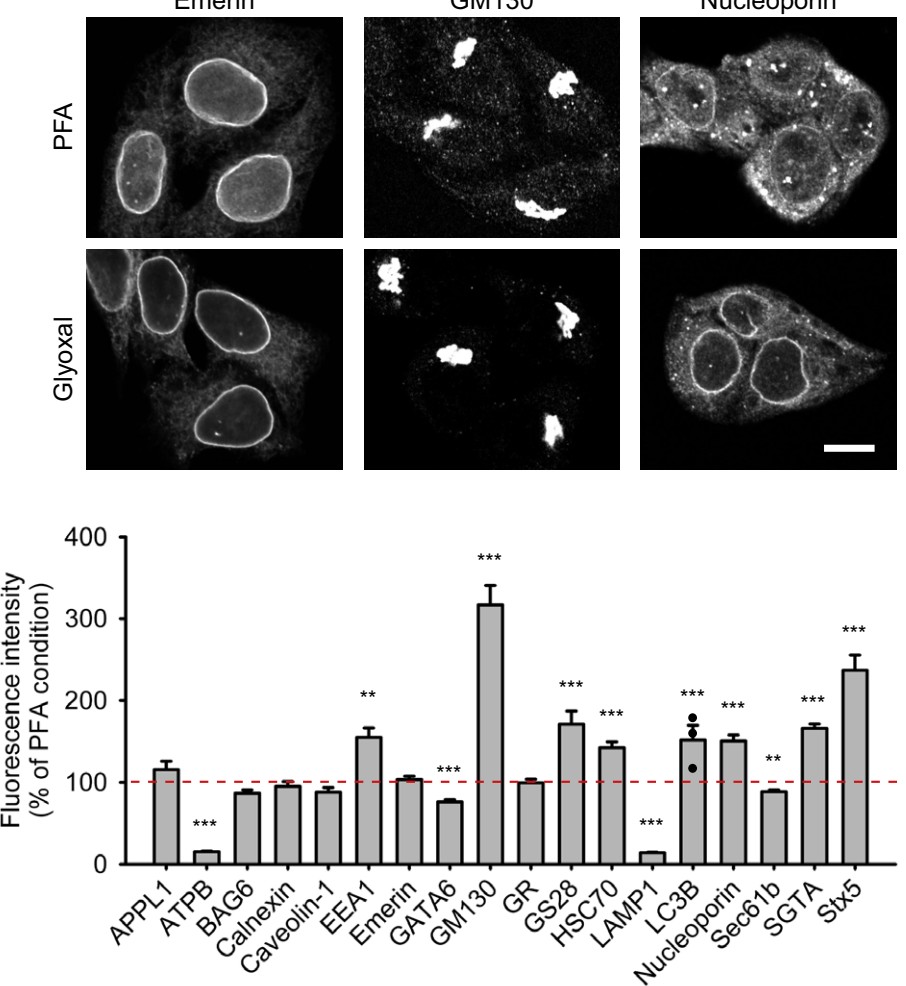

**Figure 13.  Comparison of immunostained HeLa cells after fixation with either PFA or glyoxal.**
The fluorescence intensity of a variety of proteins was compared between cells fixed with PFA or glyoxal. Quantification is shown as percentage of signal derived from PFA-fixed cells. Eight out of the 18 target proteins which were stained show significantly brighter signal when fixed with glyoxal. Only four proteins show significantly reduced staining intensity. For LC3B, the intensity of less than 5 cells was quantified; therefore, single data points were plotted in addition to the bars. $N = 3$–44 cells per condition (mean ± SEM). Scale bar = 10 μm. **$P < 0.01$, ***$P < 0.001$ (Wilcoxon rank-sum test for APPL1, ATPB, BAG6, caveolin-1, GATA6 and HSC70, two-sided Student's $t$-test for all other proteins).

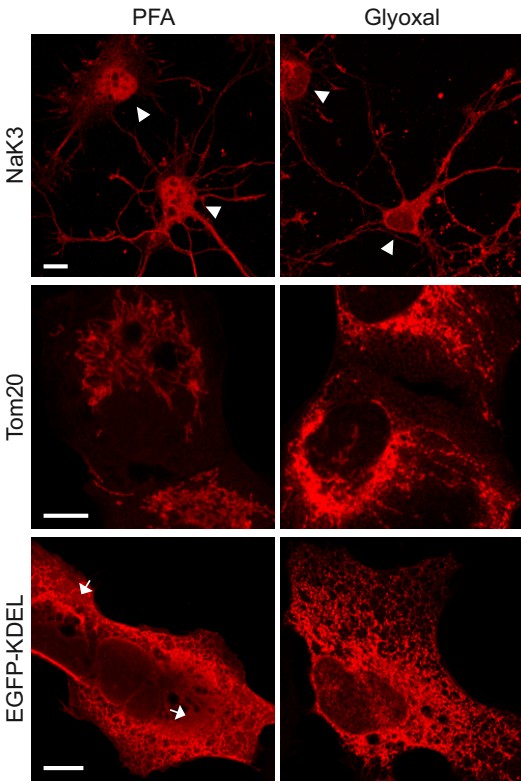

**Figure 14.  Comparison of immunostained U2OS cells and primary hippocampal neurons after either PFA or glyoxal fixation.**

Immunostaining of the Na/K ATPase in primary hippocampal neurons shows a different distribution of the protein between PFA-fixed and glyoxal-fixed samples. While in PFA-fixed neurons, the antibody falsely stains the nucleus as well as the cytoplasm (100% of the 82 cells we analyzed), in glyoxal-fixed neurons the nucleus is devoid of signal, and the membrane appears to be correctly labeled (arrowheads; 100% of the 60 cells we analyzed). U2OS cells were immunostained for mitochondria (Tom20) and ER (EGFP-KDEL). The fixation and/or staining of mitochondria seems to be comparable in glyoxal and in PFA-fixed cells. The staining of the ER shows an improved signal-to-noise ratio. The signal appears de-localized from the ER for multiple PFA-stained cells (25% of 36 analyzed cells, see arrows), while this is rare for the glyoxal-stained cells (3.4% of 58 analyzed cells). Scale bar = 10 μm.

for the presynaptic active zone protein CtBP2, the calcium buffer protein calretinin, and the postsynaptic active zone protein Homer 1. For none of the tested proteins was the signal found to be poorer in glyoxal in this preparation.

The Outeiro laboratory (Department of Experimental Neurodegeneration, Center for Nanoscale Microscopy and Molecular Physiology of the Brain, University Medical Center Göttingen; Max-Planck-Institute for Experimental Medicine, Göttingen, Germany) investigated the cytoskeletal protein vimentin, along with alpha-synuclein, a soluble protein whose propensity to aggregate is a potential cause for Parkinson's disease (Fig 11). Vimentin immunostainings were ~20% brighter after PFA fixation. A stronger phenotype was observed for alpha-synuclein immunostainings, which were twofold brighter after glyoxal fixation. The Outeiro laboratory also tested a fluorescence tag, mOrange2, in the two fixation conditions, and found that ~20% more fluorescence could be detected after glyoxal fixation.

The Rehling laboratory (Department of Cellular Biochemistry, University Medical Center Göttingen; Max-Planck-Institute for Biophysical Chemistry, Göttingen, Germany) analyzed several mitochondrial proteins in cell cultures, using conventional immunostainings, and found that the signals were better, albeit within the same range, for PFA fixations of TIM23, COA6, and NDUFA9. For ATP5B and for MitoTracker stainings, the situation was inversed, with glyoxal fixations providing substantially improved images (Fig 12).

The Schwappach laboratory (Department of Molecular Biology, University Medical Center Göttingen, Germany) also relied on cell cultures to analyze a large palette of proteins involved in several processes, from nuclear organization to mitochondria and to the secretory pathway. Seven proteins were similar for PFA and glyoxal fixations (with marginally dimmer staining for Sec61b, an endoplasmic reticulum protein, after glyoxal fixation). The immunostainings for the lysosomal marker LAMP1 and for ATPB, a component of the mitochondrial ATP synthase, were far poorer after glyoxal fixation than after PFA fixation. In contrast, eight target proteins provided brighter immunostainings after glyoxal fixation, with the largest differences seen for the Golgi marker GM130 and for the SNARE syntaxin 5 (Fig 13).

The Testa (Department of Applied Physics, KTH Royal Institute of Technology, Stockholm, Sweden) laboratory analyzed, using

**Figure 15.  Comparison of mouse tissue staining following either PFA or glyoxal fixation.**

A, B   Confocal images showing staining of olfactory marker protein (OMP) and β3-tubulin along the dorsal aspect of the mouse olfactory epithelium. While sections from both types of fixative show OMP signal in the olfactory sensory neuron somata, their dendrites, and axons, the axon bundles (green arrow) located above the olfactory epithelium exemplify the clear signal-to-noise ratio benefits of glyoxal fixation versus that of the PFA-fixative. Immunostaining with the β3-tubulin antibody stains the dendrites and axons (blue arrowheads) in both PFA and glyoxal-fixed tissue, but strong staining of the cilia (blue arrows) can only be observed in the glyoxal-fixed sections (B).

C, D   Confocal images depicting bundles of axons belonging to olfactory sensory neurons on the path toward the olfactory bulb. Identities of the axons are in part defined by the neuropilin-1 (Nrp1) and neuropilin-2 (Nrp2) expression levels, visualized here with antibodies raised against the two proteins. While complementary expression of the two molecules can be seen in the PFA-fixed sections (C), the glyoxal-fixed sections (D) exhibit profoundly improved signal-to-noise ratios for Nrp-1 (red arrows), and in the case of Nrp-2, also the segmentation of the axon bundle into varying levels of Nrp-2 (green arrows).

E, F   Confocal images of olfactory sensory neuron axons coalescing into glomeruli where they synapse with dendrites of olfactory bulb neurons. The axons of olfactory sensory neurons can be readily visualized with OMP staining (green) in the superficial olfactory nerve layer and terminating in glomeruli located below (green arrows). While sections fixed with either PFA or glyoxal display adequate staining levels, the signal distribution of the PFA-fixed tissue appears more irregular (E), seemingly lacking the neurofilamentary morphology that appears preserved in the glyoxal-fixed sections (F). The glomeruli themselves are neuropil structures comprised primarily of olfactory sensory neurons forming synapses with dendrites of mitral/tufted cells as well as dendrites of periglomerular neurons. Immunostaining with vesicular glutamate transporter 2 (VGLUT2) allows visualization of these structures and is easily seen in the glyoxal-fixed section (F), while it appears the antigen was masked by PFA fixation as no signal above background can be seen in the PFA-fixed panel (red arrows in E). Note that a different polyclonal antibody for VGLUT2 from the same provider does provide signal with PFA, albeit weaker versus glyoxal (inset). Staining with β3-tubulin touts the benefits of glyoxal both due to the signal improvements in the case of the mild staining in the axons of the olfactory nerve layer that is only visible in the glyoxal-fixed tissue, but also in preserving tissue morphology as demonstrated by the dendritic processes inside glomeruli (blue arrows) and in the external plexiform layer located below the glomeruli.

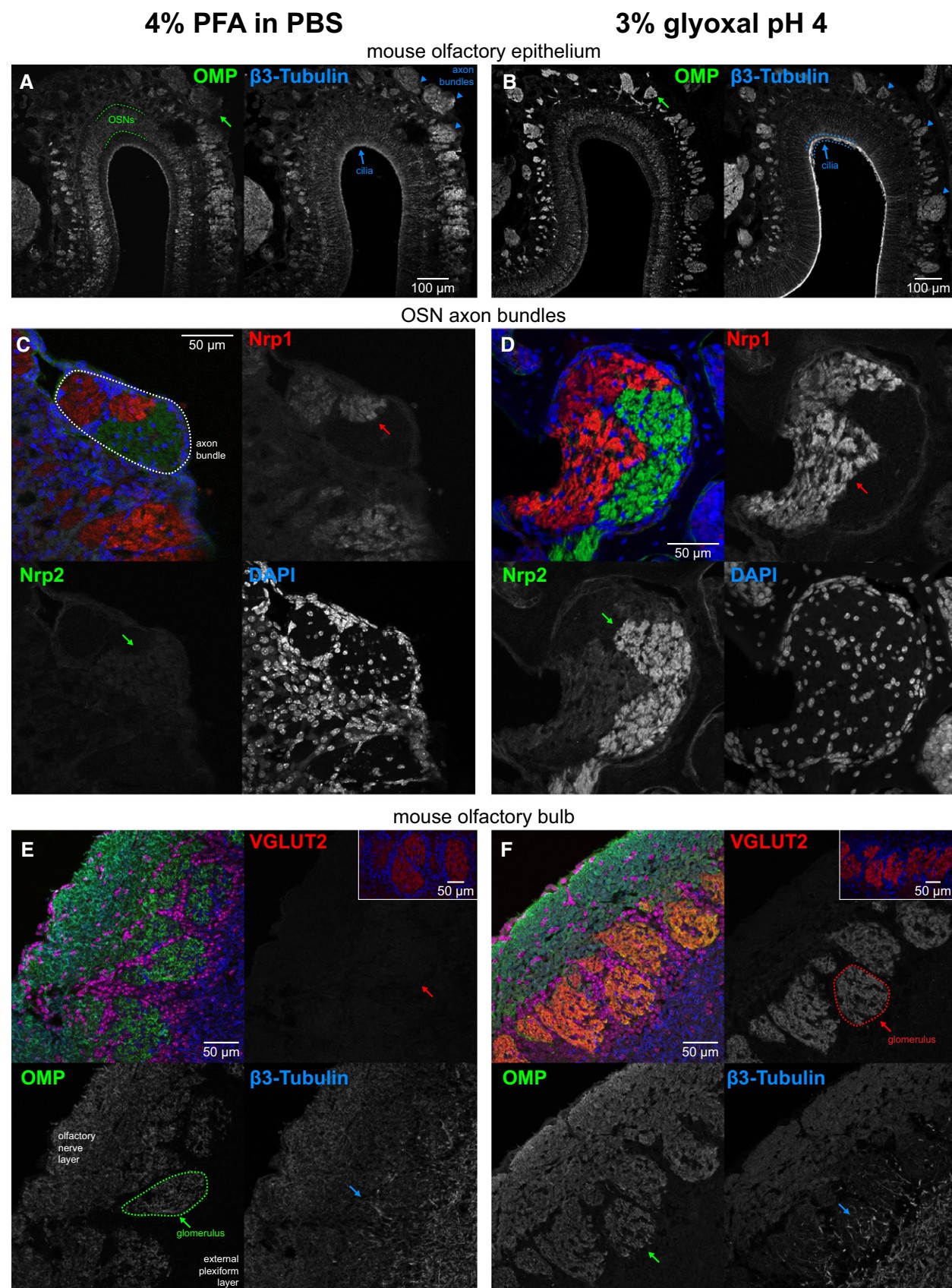

**Figure 15.**

confocal microscopy, a membrane marker (the Na$^+$/K$^+$-ATPase), a mitochondria marker (Tom20), and an endoplasmic reticulum marker (overexpressed EGFP coupled to a KDEL signal) in primary cultured neurons or in cultured cells (Fig 14). The Na$^+$/K$^+$-ATPase was revealed correctly as a membrane protein only after glyoxal fixation, while it was found mostly in the nucleus after PFA fixation. Tom20 stainings were similar for the two fixations. EGFP-KDEL stainings had a poorer morphology after PFA fixation, with this protein apparently having spilled over from the endoplasmic reticulum during fixation in a quarter of all analyzed cells.

Finally, the Zapiec group (Max Planck Research Unit for Neurogenetics, Frankfurt am Main, Germany) analyzed different proteins in the mouse olfactory epithelium and bulb and found substantially stronger immunostainings after glyoxal fixation for the olfactory marker protein (OMP; Fig 15A and B), for neuropilin-1 and neuropilin-2 (Fig 15C and D), and for the vesicular glutamate transporter 2 (Fig 15E and F). The same was observed for β3-tubulin immunostainings (Fig 15A, B and E, F).

## Discussion

We conclude that glyoxal fixation appears to be more efficient than PFA for many laboratories, in several countries. An

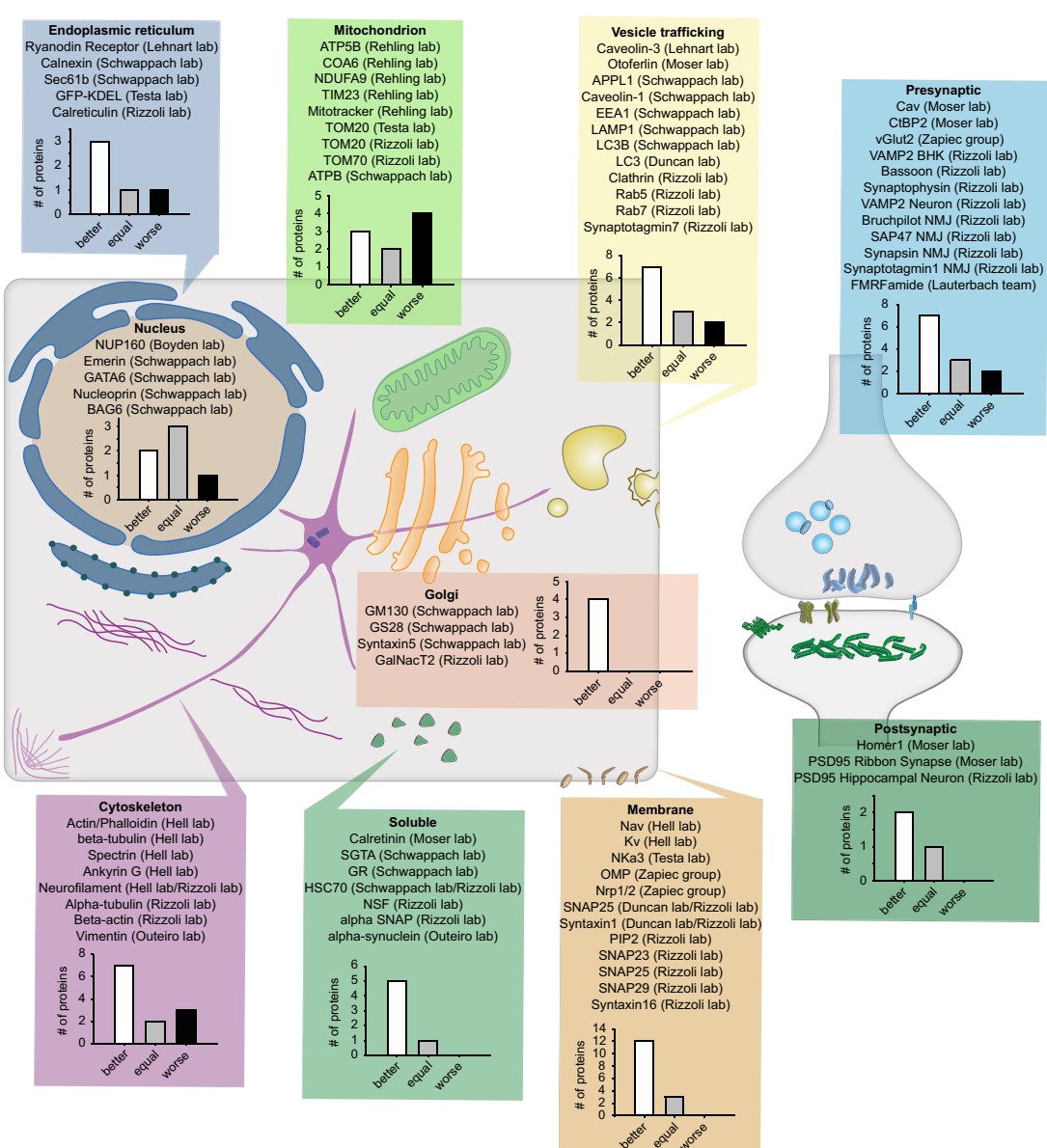

**Figure 16. Overview of the results obtained from all immunostainings, in all of the laboratories testing glyoxal.**
Various cellular targets, ranging from the nucleus to synapses of hippocampal neurons, were tested after fixation with either PFA or glyoxal by us and 11 additional laboratories. Overall, 51 targets were better stained after glyoxal fixation than after PFA fixation, 12 targets were stained worse, and 19 targets were equally well stained.

overview of the results, indicating the different cellular targets analyzed, is presented in Fig 16. Overall, 51 targets were better stained after glyoxal fixation, 12 targets were stained worse, and 19 targets were stained equally well, which implies that glyoxal fixation seems to be generally superior to PFA. The strongest difference is seen for membrane proteins and for proteins of the Golgi apparatus. The organelle for which glyoxal is least successful is the mitochondrion.

In principle, glyoxal could be combined with other fixatives, including glutaraldehyde, for an even stronger fixation. However, the behavior of aldehyde fixatives is exceedingly complex, leading to many side reactions (Migneault *et al*, 2004), which renders such an experiment difficult to reproduce. In a few trials, glutaraldehyde addition to glyoxal solutions actually caused poor morphology preservation, indicating that this may not be an optimal solution. Post-fixation with glutaraldehyde, for stronger and longer-lasting sample preservation, is nevertheless possible, as we observed in the electron microscopy experiments (Appendix Fig S15; see also the respective Materials and Methods section).

Since glyoxal is substantially less harmful by inhalation than PFA (Wicks & Suntzeff, 1943), we suggest that it should replace PFA for many applications. Comparative testing will still be needed for every antibody before settling on a fixation procedure. Nonetheless, we found that glyoxal typically provides immunostainings of better quality than PFA. In the few cases in which PFA provided brighter images, the glyoxal images were nevertheless still usable, revealing structures that appeared biologically accurate, with the clear exception of the lysosome marker LAMP1 and of the mitochondrial ATP synthase (Fig 13). The opposite situation, however, was far more often encountered, especially for the membrane proteins such as the $Na^+/K^+$-ATPase (Fig 14), the SNAREs SNAP25 and syntaxin 1 (Fig 6), or multiple proteins of the mouse olfactory epithelium (Fig 15).

While an extensive discussion of why this may be the case is beyond the purpose of this work and would require an in-depth analysis of the fixation chemistry of both PFA and glyoxal, it is probable that the appearance of uniformly distributed small spots in PFA-fixed samples (Fig 4 and Appendix Discussion) is due to insufficient cross-linking of proteins. The appearance of such spots has been a concern since the initial applications of super-resolution microscopy (see, e.g., Lang & Rizzoli, 2010), which mostly revealed structures of ~70–100 nm in size. The fact that PFA only fixes about 60% of the proteins (Fig 3) implies that a large fraction of the proteins is still mobile, can change its distribution during immunostaining, and may be even lost from the samples. We assume that the faster and stronger fixation induced by glyoxal (Figs 1 and 2) plays a central role in improving the quality of the immunostainings, by maintaining the proteins in their organelle locations.

We conclude that this feature, the stronger and more accurate fixation, makes glyoxal a good candidate for the fixative of choice in immunostainings. In our opinion, glyoxal should still be preferred even for targets for which the two fixatives work equally well, because PFA presents substantially more health hazards than glyoxal during normal, routine laboratory work (Wicks & Suntzeff, 1943).

# Materials and Methods

### Paraformaldehyde (PFA) and glyoxal preparation

For all experiments, a 4% w/v paraformaldehyde (Sigma-Aldrich #P6148) solution and a 3% v/v glyoxal (Sigma-Aldrich #128465) solution were used. Paraformaldehyde was dissolved in PBS (137 mM NaCl, 2.7 mM KCl, 10 mM $Na_2HPO_4$, 2 mM $KH_2PO_4$, pH 7.4). The glyoxal solution was prepared according to the following protocol:

For ~4 ml glyoxal solution mix:
2.835 ml $ddH_2O$
0.789 ml ethanol (absolute, for analysis)
0.313 ml glyoxal (40% stock solution from Sigma-Aldrich, #128465)
0.03 ml acetic acid

Vortex the solution and bring the pH to 4 or 5 by adding drops of 1 M NaOH until respective pH is reached. Check pH with pH indicator paper. The solution should be kept cool and used within a few days, otherwise glyoxal might precipitate. If the stock solution shows precipitation, glyoxal can be redissolved by heating the solution to ~50°C (see also information provided by Sigma-Aldrich).

Results obtained with glyoxal at pH 5 are shown in all figures, unless noted otherwise (Appendix Fig S2 shows data obtained from glyoxal pH 4). For several control experiments (as noted in the figure legends), the same amount of ethanol was added to the PFA solution.

The fixatives for the SDS–PAGE experiments (Fig 3A, Appendix Fig S7) were prepared so that the final amount of PFA and glyoxal (mixed with the cytosol samples) were 4% and 3%, respectively. As a control for the SDS–PAGE experiments, 0.2% glutaraldehyde (AppliChem #A3166) was added to a 4% PFA solution, as noted in the respective figure legend.

### Propidium iodide and FM 1-43 imaging

Measuring cell penetration by the fixative (Fig 1A and B; Appendix Fig S1) was done using the dyes propidium iodide (Sigma-Aldrich #P4170) and FM 1-43 (Biotium #70020). COS-7 fibroblast cells (obtained from the Leibniz Institute DSMZ—German Collection of Microorganisms and Cell Culture), plated on poly-L-lysine (PLL)-coated coverslips and cultured under standard conditions, were washed briefly in pre-warmed COS-7 cell Ringer (130 mM NaCl, 4 mM KCl, 5 mM $CaCl_2$, 1 mM $MgCl_2$, 48 mM glucose, 10 mM HEPES, pH 7.4). Afterward, the respective fixatives were added to the cells, containing either propidium iodide (5 μM) or FM 1-43 (1.5 μM). The cells were imaged for 60 min or 10 min, respectively, using an inverted epifluorescence microscope (Nikon Eclipse Ti-E), as described in the Imaging section, below.

To determine the intensity of the propidium iodide stainings (Fig 3B), COS-7 cells were fixed in the appropriate fixative for 30 min on ice and for another 30 min at room temperature, followed by 20 min of quenching in 100 mM $NH_4Cl$ and 100 mM glycine. After washing in PBS for 5 min, the cells were incubated in 5 μM propidium iodide in PBS for 10 min at room temperature. After a 15-min wash-off in PBS, the cells were imaged using the same microscope as in the previous paragraph.

For the optimization of glyoxal fixation (Appendix Table S1), cultured primary hippocampal neurons were fixed for 30 min on ice and another 30 min at room temperature in the respective fixative, followed by 10-min quenching in 100 mM $NH_4Cl$. The neurons were washed two times briefly in PBS and imaged in a 1.5 μM FM 1-43 solution using an Olympus IX71 inverted epifluorescence microscope described below in the imaging section.

**Fluorescence *in situ* hybridization**

Fluorescence *in situ* hybridization (Fig 3B) was performed using the QuantiGene® ViewRNA ISH Cell Assay kit (Affymetrix #QVC0001), according to the protocol provided by Affymetrix. In short, cultured rat hippocampal neurons were fixed in one of the tested fixatives for 10 min on ice and for another 20 min at room temperature. After a washing step, the cells were incubated in the provided detergent solution, followed by probe hybridization for 3 h at 40°C (using standard probes for GAPDH, provided with the kit by the manufacturer). Afterwards, the samples were washed in the provided wash buffer, and signal amplification was done by incubating the samples in pre-amplifier and amplifier solution for 30 min each at 40°C. Label hybridization was done as well for 30 min at 40°C using Cy5 as dye. After washing in wash buffer and PBS, the samples were embedded in Mowiol and imaged using an inverted Nikon Eclipse Ti-E epifluorescence microscope.

**Transferrin, LysoTracker®, and cholera toxin uptake assay**

Live imaging of transferrin (coupled to Alexa Fluor 594, Thermo Fisher #T133433) and cholera toxin subunit B (coupled to Alexa Fluor 555, Thermo Fisher #C34776) uptake during fixation (Appendix Fig S4) was done in COS-7 and HeLa (obtained from the Leibniz Institute DSMZ—German Collection of Microorganisms and Cell Culture) cells. The cells, plated on PLL-coated coverslips, were incubated in 25 μg/ml transferrin or 1 μg/ml cholera toxin at 37°C for 10 min. Afterward, the cells were washed in pre-warmed COS-7 cell Ringer and were imaged. A concentrated solution of each fixative was added to the Ringer so that the final concentration of fixative was 4% for PFA and 3% for glyoxal. The cells were imaged during the first 10 min of fixation using the inverted Nikon Eclipse Ti-E epifluorescence microscope.

The imaging of transferrin and LysoTracker uptake at different time points during fixation (Appendix Figs S2 and S3) was done in HeLa and COS-7 cells. The cells were incubated in the respective fixative for 3, 5, 10, 15 and 20 min at 37°C prior to the addition of 25 μg/ml transferrin Alexa594 or 50 nM LysoTracker Red DND-99 (Thermo Fisher #L7528). Each sample was incubated in the fixative and transferrin/LysoTracker for 20 more min. The cells were then washed with PBS and embedded in Mowiol. The samples were imaged with a confocal TCS SP5 microscope (Leica).

**Lipofectamine transfection of COS-7 cells, HeLa cells, and BHK cells**

For the imaging of preservation of various GFP-tagged proteins and structures (Appendix Figs S5 and S6), COS-7 fibroblasts or HeLa cells were transfected with a TOMM70 construct from *S. cerevisiae*, which was amplified by PCR and cloned into a pEGFP-N1 plasmid

(Clontech), as well as an EGFP-N1-α-tubulin construct, a nEGFP-N1-SNAP25 construct, a mCherry-pCS2+-GalNacT2 (which was a kind gift from Elena Taverna, MPI of Molecular Cell Biology and Genetics, Wieland Huttner group) construct, and a mOrange2-N1-synaptopHluorin construct. The chimeric mOr2-SypHy indicator was created by substituting the superecliptic GFP from the original SypHy (Granseth *et al*, 2006) construct (purchased from Addgene, Cambridge, MA, USA) with the pH-sensitive mOrange2 fluorescent protein (purchased from Addgene). One hour prior to transfection, the cells were incubated in antibiotic-free medium. Lipofectamine® 2000 (Thermo Fisher #11668) and the DNA (0.5 or 1 μg per 18-mm cover slip) were incubated in OptiMEM (Thermo Fisher #31985047) for 20 min and were subsequently added to the cells. The medium was changed back to normal culturing medium (DMEM containing fetal calf serum, glutamine, penicillin, and streptomycin) the next day, and cells were imaged using an inverted Nikon Eclipse Ti-E epifluorescence microscope. The cells were imaged in COS-7 cell Ringer before fixation and were imaged again after incubation in the different fixatives for 60 min.

For transfection with the GFP-tagged target protein VAMP2 (Appendix Fig S10), the following construct was used: pEGFP-N1-VAMP2 (backbone plasmid was purchased from Addgene). 2.5 h prior to transfection, the cells (BHK fibroblasts) were incubated in antibiotic-free medium. 1 μg of DNA per 18-mm cover slip and Lipofectamine® 2000 were incubated for 20 min in OptiMEM and afterward added to the medium. Cells were incubated in the mixture overnight and were immunostained the following day after transfection.

For SNAP-tag labeling (Appendix Fig S9), HeLa cells were transfected with the following constructs: cytoplasmatic SNAP-tag (pSNAPf, purchased from New England Biolabs), α-synuclein-SNAP-tag, VAMP2-SNAP-tag, and transferrin receptor-SNAP-tag. The SNAP-tag fused to either the N- or the C-terminal of VAMP2 was created by PCR amplification of VAMP2 (Vreja *et al*, 2015) and insertion into the SNAP-tag plasmid by Gibson assembly (Gibson *et al*, 2009). The transferrin receptor (Opazo *et al*, 2012) and α-synuclein (Lázaro *et al*, 2014) were amplified by PCR and inserted into the SNAP-tag plasmid by Gibson assembly. 1 μg of DNA per coverslip was incubated for 20 min with Lipofectamine® 2000, and 100 μg of the mixture in OptiMEM was added to each coverslip. Cells were incubated overnight, and labeling was done the following day, as described in the next section.

**SNAP-tag labeling**

Transfected HeLa cells were washed briefly in medium and then fixed with either PFA or glyoxal pH 5 for 30 min on ice and another 30 min at room temperature. The cells were labeled with 0.3 μM SNAP-Cell TMR-Star (New England BioLabs #S9105S) for 30 min and afterward washed with PBS for 10 min. TMR fluorescence was imaged at the Olympus IX71 inverted epifluorescence microscope.

**Immunocytochemistry of cultured primary hippocampal neurons**

Rat primary hippocampal neuron cultures (Fig 4 and Appendix Figs S12–S14) were prepared as described before (Opazo *et al*, 2010; Beaudoin *et al*, 2012) and were cultured either under standard conditions, or in Banker arrangements, locally separated from the

**Table 1.  The pH of glyoxal solution used for fixation of neuronal samples.**

| Staining | pH |
|---|---|
| α/β-SNAP | 4 |
| α-Tubulin | 5 |
| β-Actin | 5 |
| Bassoon | 4 |
| Calreticulin | 5 |
| Clathrin LC | 4 |
| HSC70 | 4 |
| Neurofilament L | 4 |
| NSF | 4 |
| PSD95 | 4 |
| Rab5 | 4 |
| Rab7 | 4 |
| SNAP23 | 4 |
| SNAP25 | 4 |
| SNAP29 | 4 |
| Syntaxin 1 | 5 |
| Syntaxin 16 | 4 |
| Synaptophysin | 5 |
| Synaptotagmin 7 | 4 |
| VAMP2 | 5 |

**Table 2.  Antibodies used for the immunostaining of neuronal proteins.**

| Target protein | Species | Company | Dilution |
|---|---|---|---|
| Primary antibodies | | | |
| *α/β-SNAP | Mouse | Reinhard Jahn | 1:100 |
| α-Tubulin | Rabbit | SySy (#302203) | 1:4,000 |
| β-Actin | Mouse | Sigma-Aldrich (A1978) | 1:300 |
| Bassoon | Mouse | Enzo Lifescience (#SAP7F407) | 1:100 |
| Calreticulin | Rabbit | Cell Signaling (#12238) | 1:100 |
| Clathrin LC | Mouse | SySy (#113011) | 1:1,000 |
| HSC70 | Mouse | Santa Cruz (#sc-7298) | 1:100 |
| Neurofilament L | Rabbit | SySy (#171002) | 1:500 |
| NSF | Rabbit | SySy (#123002) | 1:500 |
| PSD95 | Mouse | Neuromap (#75-028 (K28/43)) | 1:200 |
| *Rab5 | Mouse | Reinhard Jahn | 1:100 |
| Rab7 | Rabbit | Cell Signaling (#9367) | 1:100 |
| SNAP23 | Rabbit | SySy (#111202) | 1:100 |
| SNAP25 | Mouse | SySy (#111002) | 1:500 |
| SNAP29 | Rabbit | SySy (#111302) | 1:500 |
| Syntaxin 1 | Mouse | SySy (#110011) | 1:300 |
| Syntaxin 16 | Rabbit | SySy (#110162) | 1:100 |
| *Synaptophysin | Rabbit | Reinhard Jahn (G96) | 1:1,500 |
| Synaptotagmin 7 | Rabbit | SySy (#105173) | 1:100 |
| VAMP2 | Mouse | SySy(#104211) | 1:1,500 |
| Secondary antibodies | | | |
| Anti-mouse IgG (Atto647N) | Goat | Sigma-Aldrich (#50185) | 1:150 |
| Anti-rabbit IgG (Atto647N) | Goat | Rockland (#611-156-003) | 1:500 |

Indicated antibodies (*) were kind gifts of Prof. Dr. Reinhard Jahn, Max-Planck-Institute for Biophysical Chemistry, Göttingen, Germany.

astrocyte feeder layer (as described in Kaech & Banker, 2006). The neurons, plated on poly-L-lysine coated cover slips, were fixed in PFA (pH 7, pH 4/5 or with Et-OH) or glyoxal for 60 min and were subsequently quenched for 30 min in 100 mM $NH_4Cl$. The pH of the glyoxal solution used for fixation is presented in Table 1. For each antibody, we used the pH that provided a brighter immunostaining. Permeabilization and background epitope blocking were achieved by incubating the neurons for 15 min in blocking solution, containing 2.5% BSA and 0.1% Triton X-100 in PBS. The samples were incubated with primary antibodies diluted in blocking solution, for 60 min at room temperature. Table 2 presents the antibodies and their dilutions from 1 mg/ml stocks. After washing another 15 min in blocking solution, secondary antibodies were applied for 60 min, at room temperature. Subsequent washing in high-salt PBS (500 mM NaCl) and PBS was followed by embedding in Mowiol. The samples were imaged with a STED TCS SP5 microscope (Leica).

**Immunocytochemistry of HeLa and COS-7 cells**

HeLa cells that took up transferrin Alexa546 (see uptake assay described above) were immunostained for endosomes (EEA1; Appendix Fig S11). The cells were fixed in the respective fixative for 30 min on ice and another 30 min at room temperature. Afterward, they were quenched with 100 mM $NH_4Cl$ for 20 min. Permeabilization and blocking were done for 15 min in 2.5% BSA and 0.1% Triton X-100 in PBS. Subsequently, the cells were incubated in the primary antibody rabbit anti-EEA1 (Synaptic Systems #237002),

diluted 1:100 for 60 min. After washing in blocking/permeabilization solution for 15 min, the cells were incubated with the secondary antibodies for 60 min. A donkey anti-rabbit antibody coupled to Atto647N (Rockland, diluted 1:500) was used. Subsequent washing in high-salt PBS and normal PBS was followed by embedding in Mowiol, and the cells were imaged at the confocal TCS SP5 microscope (Leica).

Immunostaining of overexpressed GFP-tagged proteins (Appendix Fig S10; see transfection described earlier) was done like described above. Following primary antibodies were used: mouse anti-TOMM20 (Sigma-Aldrich #WH0009804M1), diluted 1:200, rabbit anti-α-tubulin (Synaptic Systems #302203), diluted 1:1,000,

mouse anti-VAMP2 (Synaptic Systems #104211), diluted 1:200, mouse anti-TGN38 (BD Bioscience #610898), diluted 1:100, mouse anti-SNAP25 (Synaptic Systems # 111011), diluted 1:500.

Immunostaining of phosphatidylinositol-4,5-bisphosphate ($PIP_2$) was done as described above (Appendix Fig S8). The primary antibody mouse anti-$PIP_2$ (Abcam #ab11039), diluted 1:50, was used. As secondary antibody, a donkey anti-mouse coupled to Cy2 was used in the dilution 1:100. The cells were imaged with the Olympus IX71 inverted epifluorescence microscope.

## Immunohistochemistry of *Drosophila* 3$^{rd}$-instar larvae neuromuscular junctions

*Drosophila melanogaster* 3$^{rd}$-instar larvae (Appendix Fig S16) were dissected in standard Drosophila medium as described before (Jan & Jan, 1976). The larvae were fixed for 30 min on ice, and for another 30 min at room temperature, followed by 30 min of quenching in 100 mM $NH_4Cl$. Permeabilization and blocking were performed for 30 min in PBS containing 2.5% BSA and 0.5% Triton X-100. Incubation in primary antibodies was done for 60 min at room temperature. The following antibodies were used: mouse anti-synaptotagmin 1 (3H2 2D7), diluted 1:50, mouse anti-synapsin (3C11), diluted 1:20, mouse anti-syntaxin (8C3), diluted 1:50, mouse anti-SAP47 (nc46), diluted 1:100, and mouse anti-bruchpilot (nc82), diluted 1:50. All antibodies were purchased from the Developmental Studies Hybridoma Bank at the University of Iowa (DSHB). After 30 min of washing in the blocking solution (0.5% Triton X-100), the samples were incubated in a Cy3-labeled goat anti-mouse antibody (1:100, Dianova #715-165-150) for 60 min at room temperature. Subsequently, larvae were washed in high-salt PBS and PBS and embedded in Mowiol. The samples were then imaged using an Olympus inverted epifluorescence microscope.

## Immunohistochemistry of mouse inner hair cells

Organs of Corti (Appendix Fig S17) were dissected from P14 to P18 wild-type mice in ice-cold HBSS (5.36 mM KCl, 141.7 mM NaCl, 10 mM HEPES, 34 mM L-glutamine, 6.9 mM D-glucose, 1 mM $MgCl_2$, 0.5 mM $MgSO_4$, pH 7.4). The inner hair cells were stimulated by incubating the tissue for 3 min in HBSS with high potassium (65.36 mM KCl) at 37°C. Afterward, the organs were fixed for 30 min on ice and for another 30 min at room temperature. The subsequent quenching was performed for 30 min in 100 mM $NH_4Cl$ and 100 mM glycine. The organs were then permeabilized and blocked for 30 min with PBS containing 0.5% Triton X-100 and 2.5% BSA. The primary antibodies mouse anti-otoferlin (Abcam #ab53233), diluted 1:350, and rabbit anti-ribeye (Synaptic Systems #192003), diluted 1:1,500, were applied for 60 min. After 30 min of washing, the organs were incubated in secondary antibodies for 60 min. Atto647-labeled goat anti-mouse (1:250, Sigma-Aldrich #50185) and the Cy2-labeled goat anti-rabbit (1:100, Dianova #111-225-144) secondary antibodies were used. Washing in high-salt PBS and PBS was followed by embedding in melamine, as described previously (Revelo *et al*, 2014). Organs were then cut into 200-nm thin sections using a Leica EM UC6 ultramicrotome. The sections were embedded in Mowiol and were imaged using a STED TCS SP5 microscope (Leica).

## Immunohistochemistry of mouse *levator auris longus* neuromuscular junctions

The *levator auris longus* muscle (Appendix Fig S18) was dissected from adult mice in ice-cold mouse Ringer (5 mM KCl, 154 mM NaCl, 5 mM HEPES, 11 mM D-glucose, 1 mM $MgCl_2$, 2 mM $CaCl_2$, pH 7.3). Prior to fixation, the acetylcholine receptors were stained by incubating the muscles in a 1:150 dilution of tetramethylrhodamine-labeled bungarotoxin (Sigma-Aldrich #T0195) for 15 min. After washing the tissue for 15 min in mouse Ringer, it was fixed for 30 min on ice and another 30 min at room temperature. Quenching was performed in 100 mM $NH_4Cl$ and 100 mM glycine. The tissue was then permeabilized and blocked by incubating in PBS containing 0.5% Triton X-100 and 2.5% BSA for 30 min. Primary antibodies were applied for 60 min. The following antibodies were used: mouse anti-bassoon (Enzo Lifescience #SAP7F407), diluted 1:100, and rabbit anti-piccolo (Synaptic Systems #142003), diluted 1:150. After 30 min of washing, secondary antibodies were applied for 60 min (Atto647-labeled goat anti-mouse, Sigma-Aldrich #50185, diluted 1:150, and Cy2-labeled goat anti-rabbit, Dianova #111-225-144, diluted 1:100). After 20 more min of washing in the blocking solution, 30 min in high-salt PBS, and 20 min in PBS, the samples were embedded in 2,2′-thiodiethanol as described previously (Revelo & Rizzoli, 2015; TDE, Sigma-Aldrich #166782). The neuromuscular junctions were imaged using a STED TCS SP5 microscope (Leica).

## Imaging with an inverted epifluorescence Nikon Eclipse Ti-E microscope

Experiments from Figs 1, 2 and 3B, Appendix Figs S4–S6 were imaged using the Nikon inverted epifluorescence microscope. The microscope was equipped with an HBO 100-W lamp and an IXON X3897 Andor Camera. For all samples, a 60X Plan apochromat oil immersion objective (NA 1.4) was used (from Nikon). The filter sets and time course (if applicable) used for imaging are shown in Table 3. Images were obtained using the image acquisition software NiS-Elements AR (Nikon).

## Imaging with a STED/confocal TCS SP5 microscope (Leica)

The immunostained rat hippocampal neurons (Fig 4, Appendix Figs S12–S14), mouse inner hair cells (Appendix Fig S17), and neuromuscular junctions (Appendix Fig S18), as well as the transferrin and LysoTracker uptake (Appendix Figs S2, S3 and S11) and the immunostained GFP-tagged proteins (Appendix Fig S10) were imaged using a pulsed STED microscope, built on the basis of the TCS SP5 confocal microscope (Leica). The microscope was equipped with a pulsed diode laser (18 mW, 80 MHz, 640 nm emission, PicoQuant) for excitation of the STED dye, and with a pulsed infrared titanium: sapphire (Ti:Sa) tunable laser (1W, 80 MHz, 720–1,000 nm, Mai Tai Broadband; Spectra-Physics) for depletion set at a wavelength of 750 nm. For confocal imaging, an Argon laser (488 nm) and HeNe laser lines (543, 594, 633 nm) were used for excitation. Detection was achieved by ultra-sensitive avalanche photodiodes and high sensitivity, low noise PMTs (Leica). All samples were imaged using a 100× HCX PL APO oil immersion STED objective (NA 1.4). Images were acquired using the Leica LAS AF imaging software, with a pixel

**Table 3.  Filter sets and time courses used for the Nikon Eclipse Ti-E microscope.**

| Figure panel | Excitation filter | Emission filter | Dichroic mirror | Time course |
|---|---|---|---|---|
| 1A | Cy3: 545/25 nm | 605/70 nm | 565 nm | 60 min, every 5 min |
| 1B | EGFP: 470/40 nm | 525/50 nm | 495 nm | 10 min, every 30 s |
| 2 | DIC | DIC | DIC | 60 min, every 5 min |
| 3B | Cy3: 545/25 | 605/70 nm | 565 nm | – |
| 3B | Cy5: 620/60 nm | 700/75 nm | 660 nm | – |
| Appendix Fig S5 | EGFP: 470/40 nm | 525/50 nm | 495 nm | – |
| Appendix Fig S6 (additional GFP proteins) | EGFP: 470/40 nm | 525/50 nm | 495 nm | – |
| Appendix Fig S4 | Cy3: 545/25 nm (cholera toxin) | 605/70 nm (cholera toxin) | 565 nm (cholera toxin) | 10 min, every 60 s |
|  | Texas Red: 562/40 nm (transferrin) | 624/40 nm (transferrin) | 593 nm (transferrin) |  |

size of 20 × 20 nm, 30 × 30 nm or 60 × 60 nm and a scanning speed of 1,000 Hz.

## Imaging with an inverted epifluorescence Olympus IX 71 microscope

The *Drosophila* larvae neuromuscular junctions (Appendix Fig S16), the transfected and immunostained BHK (obtained from the Max-Planck-Institute for biophysical chemistry Göttingen, Reinhard Jahn) cells (VAMP2 expression in Appendix Fig S10), the FM 1-43 stained neurons (Appendix Table S1), the COS-7 cells, stained for $PIP_2$ (Appendix Fig S8), and the SNAP-tag labeled HeLa cells (Appendix Fig S9) were imaged using an Olympus IX 71 epifluorescence microscope, equipped with a 100 W mercury lamp and a F-View II CCD camera (Soft Imaging Systems GmbH). The *Drosophila* NMJs and $PIP_2$ stained COS cells were imaged using a 100× TIRFM oil immersion objective (NA 1.45), from Olympus. The BHK cells and the SNAP-tag-labeled HeLa cells were imaged using the 40× UPlan FLN air objective (NA 0.75) from Olympus. The hippocampal neurons were imaged using a 60× UPlanApo oil immersion objective (NA 1.35) from Olympus. Filter sets used for imaging can be found in Table 4. Image acquisition was performed using the Olympus Cell$^P$ software.

## SDS–PAGE of fixed rat brain cytoplasm

Rat brain cytosol (Fig 3A and Appendix Fig S7) was prepared by homogenization of adult rat brains using a Teflon glass homogenizer in 320 mM sucrose, 5 mM HEPES, pH 7.4 (adjusted with NaOH). This was followed by a two-step centrifugation, first in an SS34 rotor (Sorvall) for 12 minutes at 14,400 g, to pellet large tissue fragments, and then in a TLA100.3 rotor (Beckman) for 60 min at 264,000 g to pellet all remaining cellular fragments. All centrifugation steps were performed at 4°C. The fixatives were prepared so that the final

**Table 4.  Filter sets used for the Olympus IX 71 epifluorescence microscope.**

| Filter | Excitation | Emission |
|---|---|---|
| FITC | 494 | 518 |
| RFP | 561 | 585 |
| Cy5 | 625 | 670 |

amount of fixative in the solution with the cytosol was 4% PFA (ph 7, pH 4, and 5) and 3% glyoxal. The samples were fixed for 15, 30, 45, or 60 min at room temperature (or 10 min at 37°C for one of the PFA fixation controls). As control samples, cytosol was also fixed with PFA plus 0.2% glutaraldehyde and PFA plus 20% ethanol. After fixation, samples were prepared for running on SDS–polyacrylamide gels by adding 2× Laemmli sample buffer (Laemmli, 1970) and heating for 5 min to 95°C. 10% polyacrylamide gels were prepared as described previously (Brunelle & Green, 2014). 25 µl of each sample and a non-fixed brain cytosol sample was run on the gels. The gels were stained in Coomassie brilliant blue overnight and were destained for 2–3 h in 50% methanol, 40% $H_2O$, 10% acetic acid the following day. The stained gels were scanned and analyzed.

## Electron microscopy

For electron microscopy of chemically fixed cells (Appendix Fig S15), primary hippocampal neurons were fixed for 20 min on ice and for another 20 min at room temperature, followed by quenching for 30 min in 100 mM $NH_4Cl$ and 100 mM glycine. The neurons were then postfixed with 2.5% glutaraldehyde for 60 min at room temperature. Another 20 min of quenching in $NH_4Cl$ and glycine were followed by 60 min of incubation in 1% osmium tetroxide. Afterward, the neurons were washed in filtered PBS for 15 min and were dehydrated with a series of ethanol dilutions. Subsequently, the cells were embedded in Epon resin by first incubating them for 3 h in a 1:1 mixture of ethanol and resin and then incubating in pure resin for 48 h at 60°C. The samples were cut into 80- to 100-nm sections using a LeicaEM UC6 ultramicrotome and were mounted on copper 50-mesh grids (Plano GmbH #2405C) or Form-var-coated copper slot grids (Plano GmbH #G2500C). The thin sections were labeled with 1% uranyl acetate for 10 min and were afterward washed for several minutes in dd$H_2O$. The samples were imaged using a JEOL JEM1011 electron microscope (JEOL GmbH), with a magnification of 10,000×.

For electron microscopy of high-pressure frozen samples (Appendix Fig S15), primary hippocampal neurons were frozen using a Leica HPM100 high-pressure freezer, using PBS with 20% polyvinylpyrrolidone as filler solution. The samples were freeze-substituted as described before (McDonald & Webb, 2011). Post-fixation was done in a mixture of 1% glutaraldehyde, 1% $OsO_4$, and 1% $H_2O$ (modified after Jiménez *et al*, 2006) prior to embedding in Epon

via an Epon dilution series (McDonald & Webb, 2011). The samples were cut into ultrathin sections (60 nm), stained in 1% uranyl acetate, and imaged with a Zeiss transmission electron microscope.

## Data analysis

All data analyses were performed automatically or semi-automatically using MATLAB (The MathWorks, Inc.), with exception of the analysis from Appendix Fig S12. Analyses in Figs 1A and B, and 3B, and in Appendix Figs S1, S2, S3, S8, S9, and S16 were performed using custom-written MATLAB routines that measure the average fluorescence intensity in manually selected regions. For Fig 1A and B, and Appendix Figs S1 and S16, the regions were selected manually. For Fig 3B, a MATLAB routine was used to separate cells from each other, using the watershed transform, and to thus determine the cellular regions of interest.

The fluorescence signals of the GFP and of the immunostainings in Appendix Fig S10 were measured by a MATLAB automatic routine that first identified the GFP signals, by applying a threshold to remove background signals, and then measured the intensity of the immunostainings in the GFP-positive regions of interest. For all analyses of the signal intensity in terms of "signal over background", signal- and background-containing regions of interest were manually determined, before dividing the average intensity in the former by the average intensity in the latter.

The analysis of the DIC images in Fig 2 was performed using a MATLAB routine that calculated the correlation coefficients of circular regions of interest (~500 nm in diameter), selected manually in the first image, to every other image taken throughout the 60 min of imaging. A similar analysis was performed for the fluorescent images from Appendix Fig S4, using circular regions of interest centered on particular organelles, selected by the user. Again, the same analysis was performed for the GFP images, before and after fixation, from Appendix Fig S6, and for the images of transferrin-labeled and immunostained cells (Appendix Fig S11). The SDS–PAGE gels in Fig 3A and Appendix Fig S7 were analyzed by measuring the overall band intensity that is left after fixation compared to the non-fixed sample. The entire length of the lanes was measured, and the intensity was summed over all bands. To avoid the smear induced by fixed molecules, which is especially evident in glutaraldehyde fixation, the signal along the lanes was first subjected to a high-pass filter.

The efficiency of preserving mitochondria during fixation (Appendix Fig S5) was analyzed by measuring the lengths of mitochondria before and after fixation. Regions of interest containing mitochondria were manually selected, and the mitochondria were detected by a thresholding procedure. The mitochondria length was then determined automatically.

For the analysis of the electron microscopy images (Appendix Fig S15), synaptic vesicles were selected manually, and line scans were applied to each vesicle.

For the analysis of the immunostained proteins in hippocampal neurons (Appendix Figs S12–S14), structures that appeared to be of organellar organization were identified and counted manually. This analysis was done blinded, randomizing both the order and the nature of the images. The number of objects was counted per immunostained $\mu m^2$, in order to take into account the different amounts of neuronal structures per image.

To analyze the structure of the observed objects, 100 typical objects were selected by an experienced observer, again in a blind fashion. The objects were clicked on, to select the center of the area. Square regions of interest, of several μm in width, were automatically generated, centered on the selected objects, and were preserved for further analysis. After all objects were selected, the regions of interest were overlaid, and each was rotated in turn (in 5° increments, using both the real image and a mirrored image), until the best possible alignment to the other regions of interest was obtained. Only the area within 1 μm from the region of interest center was used in measuring the alignments, to restrict the alignment analysis to the selected object, and not to other objects that may have been present in the regions of interest. The strength of the alignment was verified by calculating the Pearson's correlation coefficient at every angle. Once a best fit was found (with the maximal Pearson's coefficient), all images were summed, and the average object was thus obtained (shown in Appendix Figs S13 and S14). Line scans, obtained by drawing horizontal lines through the individual typical objects (after rotation), are shown in the graphs in these figures (in the form of mean ± SEM of all 100 line scans through the 100 typical objects).

## Statistics

Typically measurements were performed over multiple cells and experiments. For experiments studying multiple cells, such as neuronal immunostainings, we typically used at least 10 individual neurons in each analysis. For experiments involving single cells (such as time series obtained on one cell), we performed at least three independent experiments. For biochemical experiments, multiple experiments were performed (2–7). The sample numbers were increased if substantial variation was noted in the initial experiments. All graphs depicted here were generated using Sigma Plot (Systat Software, Inc). All bar graphs show mean values, and all error bars represent the standard error of the mean (SEM), calculated in Sigma Plot (except for the quantification of cardiomyocyte stainings in Fig 9, which represents mean values with standard deviation values). For statistical analyses in Fig 3A and Appendix Fig S16 (multiple comparisons), an one-way ANOVA with a *post hoc* Tukey test was performed. For all other statistical analyses, the two-sided Student's *t*-test (unpaired) or Wilcoxon rank-sum test was applied to the data using the in-built function in Excel or using MATLAB. For Fig 1 and Appendix Figs S4 and S5, the number of independent experiments tested (*N*) was below 5. The *t*-test was chosen, assuming that the results come from a normal distribution. The justification for this assumption is that the variation between experiments is solely driven by experimenter (pipetting) errors, which are considered to be normally distributed. For larger data sets, we used the Jarque–Bera test to verify the normal distribution of the data points. If the Jarque–Bera tests indicated normal distributions, we used *t*-tests for verifying differences between the samples. If one or both of the distributions were different from the normal distribution, according to the Jarque–Bera tests, we used a two-sample Wilcoxon rank-sum test to verify differences between the samples.

For display purposes, images were adjusted in brightness and contrast using ImageJ (Wayne Rasband, US National Institutes of Health). If intensities were compared, image adjustments in brightness and contrast were equally applied to all conditions.

     

## Animals

P14 to P18 and adult wild-type mice (*Mus musculus*) from the substrain C57Bl/6J were obtained from the University Medical Center Göttingen. Newborn wild-type Wistar rats (*Rattus norvegicus*) for the preparation of primary hippocampal neuron cultures were obtained from the University Medical Center Göttingen as well. *Drosophila melanogaster* of the Canton S strain were maintained in the laboratory, using conventional methods.

All animals were handled according to the specifications of the University of Göttingen and of the local authority, the State of Lower Saxony (Landesamt für Verbraucherschutz, LAVES, Braunschweig, Germany).

Methods of collaborating labs can be found in the Appendix.

Expanded View for this article is available online.

## Acknowledgements

We thank Katharina Grewe and Christina Schäfer for technical assistance, as well as Sebastian Jähne and Verena Klüver for experimental assistance. We thank Prof. Reinhard Jahn (Max-Planck-Institute for Biophysical Chemistry, Göttingen, Germany) for providing antibodies. This work was funded by grants from the Deutsche Forschungsgemeinschaft through the Collaborative Research Center 889 (to S.O. Rizzoli and T. Moser), the Collaborative Research Center 1002 (to S.E. Lehnart), and the Collaborative Research Center 1190 (to S.O. Rizzoli), and by a Consolidator Grant from the European Research Council (NeuroMolAnatomy ERC-CoG-614765, to S.O. Rizzoli). This work was further supported by ERC (ERCAdG 339580) to P. R. and Boehringer Ingelheim Fonds to F. R. Additional funding was provided by ITN TAMPting network to J.C.V. and B.S. (funded by the People Programme (Marie Curie Actions) of the European Union's Seventh Framework Programme (FP7/2007-2013) under the Research Executive Agency [grant number 607072]). C.V. is a fellow of the Elisabeth and Helmut Uhl Foundation. G.C. and I.T. are funded by an ERC Starting Grant 2014, 638314. Furthermore, N.H.R. is funded by an EMBO Long-Term Fellowship (ALTF 232-2016) and a Veni grant from the Netherlands Organization for Scientific Research (NWO-ALW 016.Veni.171.097). S.E.L. and E.W. are funded by the German Center for Cardiovascular Research, DZHK. E.S.B. is funded by John Doerr, the Open Philanthropy project, the HHMI-Simons Faculty Scholars Program, and US-Israel Binational Science Foundation Grant 2014509.

## Author contributions

SOR, NHR, and KNR designed the experiments. NHR and KNR performed the experiments with exception of the experiment in Appendix Fig S16, which was performed by KJS and several immunostainings of Fig 4 and Appendix Figs S12–S14, which were performed by MSH. SOR and KNR wrote the initial manuscript draft. All other authors designed and performed the experiments from Figs 5 to 15. All authors commented on and refined the manuscript. SOR supervised the project.

## Conflict of interest

The authors declare that they have no conflict of interest.

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
