## [Review Process File · The EMBO Journal]

Manuscript EMBO-2016-95709

Glyoxal as an alternative fixative to formaldehyde in immunostaining and super-resolution microscopy

Katharina N Richter, Natalia H Revelo, Katharina J Seitz, Martin S Helm, Deblina Sarkar, Rebecca S Saleeb, Elisa D'Este, Jessica Eberle, Eva Wagner, Christian Vogl, Diana F Lazaro, Frank Richter, Javier Coy-Vergara, Giovanna Coceano, Edward S Boyden, Rory R Duncan, Stefan W Hell, Marcel A Lauterbach, Stephan E Lehnart, Tobias Moser, Tiago Outeiro, Peter Rehling, Blanche Schwappach, Ilaria Testa, Bolek Zapiec, Silvio O Rizzoli

Corresponding author: Silvio Rizzoli, University of Göttingen Medical Center

Review timeline:

Submission date:	15 September 2016
Editorial Decision:	18 October 2016
Revision received:	14 January 2017
Editorial Decision:	13 February 2017
Additional Correspondence	21 February 2017
Revision received:	18 August 2017
Editorial Decision:	12 September 2017
Revision received:	25 September 2017
Accepted:	06 October 2017

Editor: Karin Dumstrei

Transaction Report:

1st Editorial Decision

18 October 2016

Thank you for submitting your manuscript to The EMBO Journal. Three referees have now evaluated your study and their comments are provided below.

As you can see from the comments there is an interest in the analysis. However, further work is also needed to conclusively demonstrate that Glyoxal is superior to PFA. In particular referee #2 is not yet convinced that the major claim is sufficiently supported by the data provided and this referee along with the others raise a number of important and valid points. Should you be able to strengthen the dataset and to address the concerns raised in full then I am interested in considering a revised version. I should add that it is EMBO Journal policy to allow only a single round of revision and that it is therefore important to resolve the raised concerns at this stage.

Let me know if we need to discuss anything further.

REFeree REPORTS

Referee #1:

In this technical report Richter et al. suggest glyoxal fixation of cellular samples to be a superior alternative to conventional fixation with 4% PFA, both with respect to speed of fixation and preservation of cellular nanostructures. They show glyoxal to fix cells more rapidly compared to PFA, while preserving cytosolic and membrane-bound antigens comparably well when imaged by either epifluorescence or STED-based superresolution microscopy. Moreover, based on SDS PAGE glyoxal provides improved fixation, in particular for cytosolic proteins while rendering nucleic acids amenable to detection by specific probes. Similar data are obtained in other preparations such as *Drosophila* larval or mouse NMJs or cochlear inner hair cells in tissue slices. Overall, the data support the authors notion that glyoxal may be superior to PFA at least when it comes to fixation of cells and possibly tissues.

This is a carefully executed technical study that may prove an important advance when it comes to fixation and preservation of antigens for fluorescence microscopy used by almost all labs in molecular and cellular biology.

I have a few technical concerns as well as some questions that should be addressed in a revised Ms.

1. A key issue is the question of fixation time that also seems critical when comparing different fixatives. The authors use plasma membrane permeability (to propidium iodide and FM1-43) as a surrogate measure of protein fixation. While I concur that most likely plasma membrane permeation relates to fixation I would like to encourage the authors to verify this by other methods. One possibility would be to monitor organelle motility (e.g. of mito-GFP), CME of transferrin and/ or maintenance of the acidic lumen of the lysosome as additional measures to monitor the kinetics of PFA vs. glyoxal fixation. Moreover, it would be interesting to determine the amount of fixed protein over time using the simple SDS-PAGE assay shown in Fig. 3 to corroborate this key point.

2. Another important point pertains to the question (see also minor points below) whether or not fixation via glyoxal more closely resembles the true distribution of protein antigens and organelles compared to PFA fixed samples. To rigorously test this it would be important to monitor the distribution of select intracellular organelles in live cells expressing fluorescent protein tagged antigens and then during the time course of fixation when under conventional PFA fixation conditions many organelles such as endosomes tend to collapse and cluster around the nucleus while being dispersed in the living cell.

3. The authors have analyzed protein antigens as well as nucleic acids. How about the preservation of lipid antigens by glyoxal vs. PFA, which except for amine lipids, may not be readily fixed but get immobilized through surrounding proteins? This would be an interesting extension and of possible importance for the lipid community.

Minor points:

Many times the Ms contains qualitative statements that are not really backed by data. For example, on p.3/ bottom it is said that the antigens analyzed by STED imaging in Fig.4 "appeared much closer to the expected distribution of the proteins". In the absence of any other method this statement seems unwarranted unless the "expected distribution" is qualified by some independent means (see my major point #2).

I suggest that images of the entire cell are presented in Figs. 2 and S2.

Referee #2:

The manuscript "glyoxal is superior to PFA in immunostaining and nanoscopy" investigates the use of glyoxal instead of PFA for cell fixation in microscopy. As the limitations of PFA as a fixative are well known, an alternative and better fixation reagent would be very useful. The manuscript contains a large number of experiment that thoroughly characterize glyoxal, and is well written. However, in my opinion, the experiments do not justify the main claim of the manuscript that glyoxal is superior to PFA:

1. The conclusion that glyoxal is superior to PFA is largely based on a visual comparison of single

images (Figure 4-7) or some rather arbitrary measurements (number of objects/um) which do not directly correlate with structure preservation. The authors repeatedly refer to "expected structures" without explaining what these might look like, or providing any reference to support that. For most images it is not apparent which fixation protocol is better, and a more quantitative and less subjective quantification of structure preservation would be very useful.

With the exception of Sup. Fig 2, where cells have been imaged before and after fixation, the authors do not show data about the ground truth structures of the proteins they imaged. This precludes any rigorous judgment whether the PFA fixed, glyoxal fixed or neither of the two structures faithfully represent the actual cellular state. Without providing this control data for the imaged structures, e.g. by live-cell fluorescence using GFP fusions or at least glutaraldehyde fixation, the authors' bold claim that glyoxal fixation is better than PFA fixation is unsubstantiated at this point.

2. An experiment I would like to see is an extension of Sup. Fig. 2. The authors should take diffraction-limited measurements of several GFP-labeled proteins before and after fixation, and directly quantify the similarity using for example an image cross-correlation analysis. In addition, they should quantify if the GFP intensities change during fixation. This parameter is important to judge epitope accessibility in Sup. Fig. 5, where intensities were normalized to the GFP signal.

3. In the literature (e.g. Dempsey 2011, and in our own hands) PFA usually results in substantially better structures for microtubules and actin (Figure 4), raising the question if a non-optimal protocol has been used here. Also, large-scale changes of cell morphology after more than 30 minutes are not regularly observed with PFA fixation.

4. An important aspect of the manuscript is that glyoxal acts much faster than PFA. However, comparing glyoxal with ethanol in the buffer with PFA without ethanol rather tests the effect on ethanol on fixation speeds. Sup. Fig. 1 even suggests that PFA + ethanol is faster than glyoxal + ethanol, thus directly contradicting the statement that glyoxal is faster than PFA.

5. The analysis of Figure 6 and 7 is in my view a bit naïve. It is not clear what averages of inhomogeneous structures should look like, and thus they are not well suited to compare the two fixation conditions. I would suggest removing these images altogether or at least using them only as supplementary figures.

Minor comments:

1. The manuscript contains a lot of data, but many figures show only very slight effects and do not necessarily support the conclusions (e.g. Fig. 6/7, Sup. Fig. 4, 6, 7, 8, 10, 11). For readability, the authors could consider reducing the number of figures, focusing on the important ones.
2. For general interest, the authors could test preservation of the activity of an enzymatic labeling tag (e.g. Halo, Snap-tag) under glyoxal fixation.
3. How exactly was the quantification made in the SDS gel in figure 3 and Sup. fig 3? Please indicate how and which bands were quantified. PFA with and without ethanol look very different on the gel, especially due to the strong band at the top in PFA + ethanol, but are not significantly different in the quantification.
4. The authors state /conclude that 40 % of the proteins remain unfixed in PFA fixed cells. In Fig 3 only total protein mass was quantified. Thus, while 40 % of all protein mass might be unfixed, judging from the low number of bands most likely substantially more than 60 % of all proteins are fixed by PFA.

Referee #3:

The authors compared Glyoxal+ethanol and standard PFA fixations mainly using cultured neurons. The authors nicely demonstrated that glyoxal provides a fixative of choice for future experiments in standard and high-resolution microscopy using several model system including cell line, drosophila and mouse tissues. Fixation is faster and subcellular structures or organelles are better persevered. I have some minor comments below. Fixation and microscopy are critical in most fields of biology, the protocol described in the manuscript merits publication in EMBO.

Minor comments:

- Please provide intermediate time points for Fig. 1 to better demonstrate the rapid penetration of glyoxal.
- While the addition of ethanol does not improve preservation of mitochondria in PFA fixation, does it improve preservation/staining of other cytoplasmic components ?
- "Glyoxal (both pH 4 and 5) reduced this unfixed pool to ~20%, relatively close to the value obtained with a mixture of PFA and glutaraldehyde, which is known to result in strong fixation² (10-15%)." This is clearly an overstatement based on the gel and the quantification. PFA+Glu is almost twice better.
- Supplementary Fig 4: "we performed line scans through individual protein domains. No substantial differences could be noted, albeit the glyoxal-fixed samples appeared brighter, as expected from the higher fixation efficiency of this compound, which would lead to a stronger retention of proteins on the membranes (Fig. 3)". The statement is not supported by the figure.
- "-actin and -tubulin, as well as neurofilaments, were better preserved and less fragmented in glyoxal-fixed neurons. ". Please quantify the fragmentation. Please correct the sentence as "actin filament" and "microtubule" are less fragmented not the monomers.
- the procedure used to align and average objects need be fully described in M&M, citing Ref 22 is not sufficient.

1st Revision - authors' response

14 January 2017

Referee #1:

In this technical report Richter et al. suggest glyoxal fixation of cellular samples to be a superior alternative to conventional fixation with 4% PFA, both with respect to speed of fixation and preservation of cellular nanostructures. They show glyoxal to fix cells more rapidly compared to PFA, while preserving cytosolic and membrane-bound antigens comparably well when imaged by either epifluorescence or STED-based superresolution microscopy. Moreover, based on SDS PAGE glyoxal provides improved fixation, in particular for cytosolic proteins while rendering nucleic acids amenable to detection by specific probes. Similar data are obtained in other preparations such as Drosophila larval or mouse NMJs or cochlear inner hair cells in tissue slices. Overall, the data support the authors notion that glyoxal may be superior to PFA at least when it comes to fixation of cells and possibly tissues.

This is a carefully executed technical study that may prove an important advance when it comes to fixation and preservation of antigens for fluorescence microscopy used by almost all labs in molecular and cellular biology.

We thank the referee for the comments.

I have a few technical concerns as well as some questions that should be addressed in a revised Ms. 1. A key issue is the question of fixation time that also seems critical when comparing different fixatives. The authors use plasma membrane permeability (to propidium iodide and FM1-43) as a surrogate measure of protein fixation. While I concur that most likely plasma membrane permeation relates to fixation I would like to encourage the authors to verify this by other methods. One possibility would be to monitor organelle motility (e.g. of mito-GFP), CME of transferrin and/ or maintenance of the acidic lumen of the lysosome as additional measures to monitor the kinetics of PFA vs. glyoxal fixation.

We have performed all of the suggested experiments:

- We monitored organelle motility during fixation with either glyoxal or PFA. We imaged endosomes labeled with fluorescently labeled transferrin, or with fluorescently labeled cholera toxin. In both cases the motility was lower in glyoxal fixed samples (see the new Supplementary Fig. 4).

- We investigated transferrin endocytosis during fixation with either glyoxal or PFA. Glyoxal resulted in fixation of transferrin on the plasma membrane. PFA allowed a substantial amount of the fixed transferrin to be internalized (Supplementary Fig. 2).
- We used the lysosome-marking probe LysoTracker to determine whether the acidic lumen of the lysosome was maintained after fixation (Supplementary Fig. 3). We observed substantial LysoTracker labeling after PFA fixation, but not after glyoxal fixation (Supplementary Fig. 3), which again implies that the latter produces stronger, more complete fixation.

Moreover, it would be interesting to determine the amount of fixed protein over time using the simple SDS-PAGE assay shown in Fig. 3 to corroborate this key point.

We have used the SDS-PAGE assay to investigate the time course of fixation in more detail, as suggested by the referee. We had originally only shown only a 60' time point; we have now included 15', 30' and 45' time points. We had originally claimed that glyoxal fixed the samples more strongly than PFA (at the 60' minute time point). The observations at earlier time points fully confirmed this (Fig. 3 and the new Supplementary Fig. 7).

2. Another important point pertains to the question (see also minor points below) whether or not fixation via glyoxal more closely resembles the true distribution of protein antigens and organelles compared to PFA fixed samples. To rigorously test this it would be important to monitor the distribution of select intracellular organelles in live cells expressing fluorescent protein tagged antigens and then during the time course of fixation when under conventional PFA fixation conditions many organelles such as endosomes tend to collapse and cluster around the nucleus while being dispersed in the living cell.

We have addressed this important point by two different experiments:

- First, we performed the experiment suggested by the referee. We expressed several different GFP-tagged proteins in cultured cells, and imaged them before and after fixation via glyoxal or PFA. We used markers for mitochondria (TOMM70), the Golgi apparatus (GalNact), the plasma membrane (SNAP25), the cytoskeleton (tubulin), and for intracellular vesicles (synaptophysin). For most of these markers the behavior of the two fixatives was similar (Supplementary Fig. 6). Nevertheless, glyoxal preserved the organization of the most mobile elements we tested (the vesicles) about 50% better.
- Second, we argued that the experiment proposed by the referee only tests the morphological accuracy of fixation, but does not investigate the efficiency of the subsequent immunostaining of these structures. To test this we immunostained the cells expressing the different GFP-coupled markers, after fixation via glyoxal or PFA (Supplementary Fig. 10). We then measured the immunostaining intensity of the structures marked by the GFP fluorescence (mitochondria, Golgi apparatus, etc). In all cases the immunostaining intensity was significantly higher after glyoxal fixation than after PFA fixation.

3. The authors have analyzed protein antigens as well as nucleic acids. How about the preservation of lipid antigens by glyoxal vs. PFA, which except for amine lipids, may not be readily fixed but get immobilized through surrounding proteins? This would be an interesting extension and of possible importance for the lipid community.

To test this we immunostained cultured cells for phosphatidylinositol-(4,5)-P₂ (PIP₂). The intensity of the immunostaining was substantially higher after glyoxal fixation (the new Supplementary Fig. 8).

Minor points:

Many times the Ms contains qualitative statements that are not really backed by data.

We have corrected all such statements that we could identify.

For example, on p.3/ bottom it is said that the antigens analyzed by STED imaging in Fig.4 "appeared much closer to the expected distribution of the proteins". In the absence of any other

method this statement seems unwarranted unless the "expected distribution" is qualified by some independent means (see my major point #2).

This is a key point, which we now address in detail. First, we confirmed this statement by the experiments performed for the referee's major point #2, using GFP-coupled proteins as markers for the expected distributions. Second, to further confirm it without the use of GFP-coupled marker proteins, we immunostained cells that had been incubated with fluorescently labeled transferrin for 10 minutes (the new Supplementary Fig. 11). Based on the current literature, we expected that the transferrin label colocalize with early endosomal markers such as EEA1. This colocalization was substantially higher after glyoxal fixation (Supplementary Fig. 11), therefore confirming the expected distribution in a better fashion than PFA.

I suggest that images of the entire cell are presented in Figs. 2 and S5.

We have performed this, showing the full frames.

Referee #2:

The manuscript "glyoxal is superior to PFA in immunostaining and nanoscopy" investigates the use of glyoxal instead of PFA for cell fixation in microscopy. As the limitations of PFA as a fixative are well known, an alternative and better fixation reagent would be very useful. The manuscript contains a large number of experiment that thoroughly characterize glyoxal, and is well written.

We thank the referee for the comments.

However, in my opinion, the experiments do not justify the main claim of the manuscript that glyoxal is superior to PFA:

1. The conclusion that glyoxal is superior to PFA is largely based on a visual comparison of single images (Figure 4-7) or some rather arbitrary measurements (number of objects/um) which do not directly correlate with structure preservation. The authors repeatedly refer to "expected structures" without explaining what these might look like, or providing any reference to support that. For most images it is not apparent which fixation protocol is better, and a more quantitative and less subjective quantification of structure preservation would be very useful. With the exception of Sup. Fig 5, where cells have been imaged before and after fixation, the authors do not show data about the ground truth structures of the proteins they imaged. This precludes any rigorous judgment whether the PFA fixed, glyoxal fixed or neither of the two structures faithfully represent the actual cellular state. Without providing this control data for the imaged structures, e.g. by live-cell fluorescence using GFP fusions or at least glutaraldehyde fixation, the authors' bold claim that glyoxal fixation is better than PFA fixation is unsubstantiated at this point.

We have adjusted the entire section, to eliminate all statements that were not fully covered by the data. We have also addressed this point by several new experiments (as explained also in our replies to the first referee, major point #2 and second minor point). In brief:

- We expressed in cultured cells fluorescently tagged markers for mitochondria, the Golgi apparatus, the plasma membrane, the cytoskeleton, and for intracellular vesicles, and imaged them before and after fixation. Glyoxal fixation preserved the organization of these markers at least as well as PFA, or, as in the case of vesicles, to a much better level (the new Supplementary Fig. 6).
- We then immunostained the cells expressing the markers, and measured the amount of immunostaining in the respective cellular structures. Glyoxal fixation enabled significantly higher immunostaining for all of the structures identified by the GFP fluorescence (the new Supplementary Fig. 10).
- To also use this type of analysis without the expression of fluorescently tagged proteins, we immunostained cells that had been incubated with fluorescently labeled transferrin (Supplementary Fig. 11). The transferrin is in this case taken up by endosomes, which should colocalize with endosomal markers such as EEA1. The colocalization was substantially higher after glyoxal fixation (the new Supplementary Fig. 11), suggesting that immunostaining after glyoxal fixation represents more faithfully the native cellular state.

2. *An experiment I would like to see is an extension of Sup. Fig. 5. The authors should take diffraction-limited measurements of several GFP-labeled proteins before and after fixation, and directly quantify the similarity using for example an image cross-correlation analysis.*

We have performed the experiment, as indicated in our reply to the previous point (the new Supplementary Fig. 6). We have also analyzed the mobility of endosomes (labeled with fluorescently-coupled transferrin or cholera toxin) during fixation (the new Supplementary Fig. 4), and found that they showed lower motility during glyoxal fixation.

In addition, they should quantify if the GFP intensities change during fixation. This parameter is important to judge epitope accessibility in former Sup. Fig. 5, where intensities were normalized to the GFP signal.

The GFP intensities change indeed during fixation, but to relatively similar levels. Using glyoxal at pH 5, which is the buffer we used most commonly in our experiments, $87 \pm 9\%$ of the GFP intensity was preserved (5 independent experiments). Using normal PFA fixation, the preservation was $76 \pm 5\%$ (6 independent experiments), not significantly different from that obtained with glyoxal.

3. *In the literature (e.g. Dempsey 2011, and in our own hands) PFA usually results in substantially better structures for microtubules and actin (Figure 4), raising the question if a non-optimal protocol has been used here.*

We assume the referee refers to the article “Evaluation of fluorophores for optimal performance in localization-based super-resolution imaging. Dempsey GT1, Vaughan JC, Chen KH, Bates M & Zhuang X; *Nat Methods* 2011 8(12):1027-36. doi: 10.1038/nmeth.1768.” The referee is wrong in assuming that this is a PFA fixation protocol. The online methods section of the article mentions: “*The immunostaining procedure for microtubules consisted of fixation for 10 min with 3% paraformaldehyde and 0.1% glutaraldehyde in PBS, washing with PBS [...]*”. Moreover, for fixation of *in vitro*-assembled microtubules the authors did the following: “*The immobilized microtubules were fixed by incubation for 10 min with 0.5% glutaraldehyde in PEM with 1 mM paclitaxel followed by washing with PEM.*”

We are aware of the fact that glutaraldehyde addition improves preservation of the cytoskeleton. At the same time, it reduces drastically immunogenicity, as we discussed in the manuscript, and is therefore not a preferred protocol for most immunostaining and imaging procedures. We have therefore compared glyoxal with PFA fixation – not with PFA mixed with glutaraldehyde, which is successful for only a handful of targets (which include, as the referee mentions, actin and tubulin).

Also, large-scale changes of cell morphology after more than 30 minutes are not regularly observed with PFA fixation.

Incubation with PFA for just 30 minutes has been shown in the past to result in the actual fixation (cross-linking) of only a small population of membrane molecules (Tanaka et al, *Nature Methods*, 2010; our original reference #2). Even 90 minutes of fixation were unable to bring the cross-linking to the levels obtained with glutaraldehyde, according to this publication. In our own hands (Supplementary Fig. 7), fixation with PFA for 30 minutes only cross-links ~45-50% of the proteins. Therefore, we are not surprised that slow changes in cell morphology continue to take place even after this time point.

4. *An important aspect of the manuscript is that glyoxal acts much faster than PFA. However, comparing glyoxal with ethanol in the buffer with PFA without ethanol rather tests the effect on ethanol on fixation speeds. Sup. Fig. 1 even suggests that PFA + ethanol is faster than glyoxal + ethanol, thus directly contradicting the statement that glyoxal is faster than PFA.*

We would like to point out that Sup. Fig. 1 only implies that ethanol addition allows rapid membrane permeation during PFA addition, but it makes no claims as to the actual fixation.

To test this we have used an SDS-PAGE assay, as in Fig. 3 of the original manuscript, and compared glyoxal (+ ethanol), PFA, and PFA + ethanol. We had originally only shown a 60-minute

fixation time point. We have now included 15, 30 and 45 minutes time points. The results support our original claims fully (Fig. 3 and the new Supplementary Fig. 7). PFA + ethanol is actually the poorest fixative for the first two time points (the one that results in the largest average amounts of unfixed proteins), implying that it is not faster than any of the other fixatives.

5. The analysis of Figure 6 and 7 is in my view a bit too naïve.

The procedure we used here, based on averaging structures from super-resolution imaging, has been used for several years in many laboratories. It is a standard procedure, which is meant to generate more information than can be observed from the single frames. Due to the spotty nature of the single super-resolution frames, they typically miss effects that become evident in the average images. This procedure has been used, for example, to show the eightfold symmetry of components of the nuclear pore complex, which was difficult to perceive in single images (Löschberger et al, *J Cell Science*, 2012). We have also used this type of analysis in the past for studying numerous proteins in synaptic boutons (for example Wilhelm et al, *Science*, 2014; our original reference #22) or in other structures (Revelo et al, *J Cell Biology*, 2014; our original reference #28).

We agree with the referee, however, that our original description of this procedure, and the interpretation of the results, was not optimally performed. We have re-written and re-organized the entire section, which should make our meaning clearer.

It is not clear what averages of inhomogeneous structures should look like, and thus they are not well suited to compare the two fixation conditions. I would suggest removing these images altogether or at least using them only as supplementary figures.

As indicated above, we have re-written the entire section, and we have also shortened it. At the same time, we would be ready to remove these images to the supplement, should we be so instructed by the editor.

Minor comments:

1. The manuscript contains a lot of data, but many figures show only very slight effects and do not necessarily support the conclusions (e.g. Fig. 6/7, former Sup. Fig. 4, 6, 7, 8, 10, 11). For readability, the authors could consider reducing the number of figures, focusing on the important ones.

We have discussed Fig. 6/7 in the previous comment. As to the other figures, we see the original Supplementary Fig. 7 (now 12), which is our only example of electron microscopy, as very important. Supplementary Figs. 10 (now 14) and 11 (now 15), which show examples of fixation and immunostaining of whole tissues, are also important for researchers using such preparations.

We agree with the referee that the original Supplementary Figs. 4, 6 and 8 were not essential, and we have removed them. In addition, we have also removed the original Supplementary Fig. 5, which has now been rendered superfluous by the Supplementary Figs. 6, 10 and 11.

2. For general interest, the authors could test preservation of the activity of an enzymatic labeling tag (e.g. Halo, Snap-tag) under glyoxal fixation.

We have performed this experiment, using the SNAP tag, expressed either alone, or coupled to three different proteins (alpha-synuclein, transferrin receptor, VAMP2). In all cases the SNAP labeling was significantly higher after glyoxal fixation (Supplementary Fig. 9).

3. How exactly was the quantification made in the SDS gel in figure 3 and Sup. fig 7? Please indicate how and which bands were quantified. PFA with and without ethanol look very different on the gel, especially due to the strong band at the top in PFA + ethanol, but are not significantly different in the quantification.

In the analysis shown in the manuscript we quantified *all* bands that survived fixation. We measured the signal intensity along the entire length of the lanes, and thus summed the intensity over all bands. To avoid the smear induced by fixed molecules, which was especially evident in

glutaraldehyde fixation, the signal along the lanes was first subjected to a high-pass filter. This allowed only the actual bands to be measured.

To address the referee's comment, we have also analyzed two randomly chosen individual bands, indicated by arrowheads in the image below. As shown in the graphs, their behavior is similar to that of the overall band analysis from the manuscript (shown in the new Supplementary Fig. 7).

Fig. 1. Single band analysis of the polyacrylamide gels from figure 3. In addition to analyzing the intensity of all bands, the intensities of two single bands (indicated in the gel image by arrowheads) were measured separately. The graphs show that, as for all bands analyzed together, glyoxal fixation was more rapid, and stronger. N = between 2 and 7 independent experiments per time point and condition.

4. The authors state /conclude that 40 % of the proteins remain unfixed in PFA fixed cells. In Fig 3 only total protein mass was quantified. Thus, while 40 % of all protein mass might be unfixed, judging from the low number of bands most likely substantially more than 60 % of all proteins are fixed by PFA.

The number of bands is not a representative statistic for this experiment. Each band is probably composed of multiple proteins in a complex sample such as the rat brain cytosol used here. Therefore, observing the disappearance of one band does not indicate that one single individual protein has been well fixed. Conversely, the presence of a band after fixation does not state that one single individual protein has remained unfixed. We therefore do not use this type of analysis.

Referee #3:

The authors compared Glyoxal+ethanol and standard PFA fixations mainly using cultured neurons. The authors nicely demonstrated that glyoxal provides a fixative of choice for future experiments in standard and high-resolution microscopy using several model system including cell line, drosophila and mouse tissues. Fixation is faster and subcellular structures or organelles are better persevered. I have some minor comments below. Fixation and microscopy are critical in most fields of biology, the protocol described in the manuscript merits publication in EMBO.

We thank the referee for the comments.

Minor comments:

- Please provide intermediate time points for Fig. 1 to better demonstrate the rapid penetration of glyoxal.

We have performed this.

- While the addition of ethanol does not improve preservation of mitochondria in PFA fixation, does it improve preservation/staining of other cytoplasmic components ?

Ethanol did not improve preservation of cytoplasmic components in our biochemical analysis of the fixation of rat brain cytosol (Fig. 3 and Supplementary Fig. 7). We therefore do not expect it to be useful in PFA fixation experiments.

- "Glyoxal (both pH 4 and 5) reduced this unfixed pool to ~20%, relatively close to the value obtained with a mixture of PFA and glutaraldehyde, which is known to result in strong fixation (10-15%)." This is clearly an overstatement based on the gel and the quantification. PFA+Glu is almost twice better.

We have now eliminated this over-statement.

- former Supplementary Fig 4: "we performed line scans through individual protein domains. No substantial differences could be noted, albeit the glyoxal-fixed samples appeared brighter, as expected from the higher fixation efficiency of this compound, which would lead to a stronger retention of proteins on the membranes (Fig. 3)." The statement is not supported by the figure.

According to the comments of the second referee, who argued that our manuscript contains too many figures, we have now removed Supplementary Fig. 4.

- "-actin and -tubulin, as well as neurofilaments, were better preserved and less fragmented in glyoxal-fixed neurons. ". Please quantify the fragmentation.

The fragmentation was quantified by the averaging performed in Fig. 6, and in the original Supplementary Fig. 8 (now removed, according to the comments of the second referee). These figures indicated that actin filaments average to strands of ~650 nm after PFA fixation, and to more than 2 μm in glyoxal fixation (where 2 μm was the maximal scale used in the measurements, not the maximal size of the filaments). The same was observed for microtubules, where PFA fixation resulted in average structures of ~250 nm length, while they were more than 2 μm in length for glyoxal fixation.

Please correct the sentence as "actin filament" and "microtubule" are less fragmented not the monomers.

We have now corrected this.

- the procedure used to align and average objects need be fully described in M&M, citing Ref 22 is not sufficient.

We have now expanded this section of the M&M substantially.

2nd Editorial Decision

13 February 2017

Thank you for submitting your revised manuscript to The EMBO Journal. Your study has now been re-reviewed by referees #1 and 2.

As you can see below while referee #1 is satisfied with the introduced revisions, referee #2 still finds that there is not enough data to support that Glyoxal is a better fixative than PFA. Given the concerns raised by referee #2, I decided to seek additional input from a new referee (referee #4) on the manuscript and the remaining concerns raised by referee #2. I am afraid that referee #4 is in agreement with referee #2 and finds that the data is not conclusive enough to support that Glyoxal is better than PFA. The referee is also not convinced that you are using an optimal PFA fixation

protocol, which makes it difficult to assess if Glyoxal is better.

Given these comments, I am afraid that I can't offer to consider publication here. I am very sorry that I can't be more positive on this occasion, but unfortunately see no other choice in this case.

REFEREE REPORTS

Referee #1:

The majority of questions raised in the initial review have been answered by adding a large amount of new data. I nonetheless have two remaining points of concern that need to be tackled before publication:

Fig. S2 is not compelling in my view. What are the images on the top? It seems as if only a fraction of a cell is shown. Moreover, internalized transferrin is expected to exhibit a punctate vesicular pattern rather than the reticular pattern seen in the PFA fixed sample. How is this explained? Finally, I do not understand why the fixative remains present during the uptake as the question to address really is whether PFA fixed cells remain endocytosis competent longer than glyoxal fixed ones.

The PIP2 stains in Fig. S8 do not live up to the standard of an EMBO J paper, irrespective of whether PFA or glyoxal is used as fixative. While publication of the Ms does not depend on this experiment I suggest that better and more compelling data using established lipid antibodies and their colocalization with PH probes are used to test preservation of lipid antigens.

Referee #2:

The revised version of the manuscript "Glyoxal is superior to PFA in immunostaining and nanoscopy" contains several additional experiments and clarifications, and is thus improved. Unfortunately, in my view, the bold claim that Glyoxal is a better fixative than PFA is not sufficiently substantiated by the data, also because for many experiments the effect is rather small or difficult to quantify and there seem to be some inconsistencies. Also, the effect is often overstated. Of course, it could be true that glyoxal is a better fixative than PFA, but if it is not, it would have negatively impact the many groups which might change their fixation protocols based on this manuscript.

Major comments:

1. Effect of fixative vs effect of buffer: As the authors use different fixation buffers (addition of Ethanol, pH), it is not clear if the perceived differences are due to the fixative or the buffer. For example, PFA+EtOH clearly shows faster fixation than glyoxal, in contrast to their statement that glyoxal is a faster fixative than PFA. Also, totally different pH values were used, which might be relevant, as also PFA might show an improved performance under low pH (eg. P 203 in: Puchtler, H., & Meloan, S. N. (1985). On the chemistry of formaldehyde fixation and its effects on immunohistochemical reactions. *Histochemistry*, 82(3), 201-204). Thus, as long as different buffers are used, I would not support the bold claim that PFA is better than glyoxal, rather the authors could call it an alternative fixation method. To make that claim, the authors have to substantiate that neither the pH, or the ethanol would make PFA as good or better than glyoxal. To address this point, the authors should repeat all experiments using the same buffer for PFA and glyoxal (also e.g. for Figure S7).

2. I not convinced that the authors use an optimal PFA protocol. In Figure 2 they observe large-scale morphological changes even after 50 minutes, and have not even reached equilibrium (I would suggest plotting the curves until an equilibrium is reached). Even the glyoxal fixed cells still change after 20 minutes. Also, most of the examples in Figure S6 show the same correlation with PFA and glyoxal, but I would expect a large scale morphological change (Figure 2) to affect the positions of all organelles and thus markers, and therefore a much more pronounced difference in S6 for all

markers. Additionally, in our own experiments when we fix on the microscope with PFA, we observe an immobilization of our structures of interest within 10-20 seconds.

Minor comments

1. Figure 5: I am not sure if objects/um² is a right measure, especially if it is not clear what is an object, and if more objects (better preservation) or fewer objects (lower fragmentation) is better. From the images, it is not clear which condition is better, as again we do not know the ground truth.
2. Figure 6/7: Averaging for superresolution microscopy might become a standard for well-defined complexes (as the NPC), but for highly heterogeneous structures, which are not regular at all, I still find it naïve, and the structures in Figure 6 are like that. Even on regular structures the averaging algorithms are not very robust and tend to overlay bright spots. Also, the averages in Figure 6 often produce very different structures than visible in the single images (Fig 4, 5). Thus, I don't think this analysis helps to compare the fixation protocols.
3. Figure S1: how can PFA with EtOH be so much faster than glyoxal, if PFA alone does not permeabilize the membrane at all? if this data is true, it would indicate that glyoxal slows down membrane permeabilization by EtOH - is that possible?
4. Figure S2: How was the membrane segmented for the PFA? Maybe transferrin does not bind any more to the PFA-fixed receptors, and just enters the cell? Were both images taken at a focal plane through the centre of the cell?
5. Figure S3: this is expected if the membranes are permeabilized by EtOH, but does characterize PFA vs glyoxal.
6. Figure S4: seems to contradict Figure 2, as clearly transferrin uptake is stopped within 1 minute.
7. Figure S5: why not use the correlation-based analysis as before? The increase in brightness upon glyoxal fixation is unexpected.
8. Figure S6: I would suggest showing the image for Synaptophysin, as this is the only protein where a difference is observed. Also, please choose similar cells for PFA and glyoxal.
9. Figure S8: As to my knowledge PIP2 does not have an amine group, probably the authors test lipid immobilization by protein fixation, not fixation of the lipids themselves.
10. Figure S9: how exactly was the intensity quantified? In the image, the glyoxal shows a diffuse bright patch. The authors should make sure that their quantification is not influenced by unspecific labelling.
11. Figure S12: please show the profile for HPF.
12. Table S1: Please include the PFA only.

Referee #4:

The authors claim to have discovered a new fixative (Glyoxal) that is less toxic than PFA and is a better preservative of cellular morphology. Unfortunately, the data shown for PFA fixation is below standard, suggesting that the authors have not optimized their fixation and/or imaging conditions in cells that are PFA fixed. Without this comparison, it is impossible to know whether or not glyoxal is superior. None of the images in this paper are of high quality. The authors must first show an example of optimally PFA fixed and stained images before I can believe that glyoxal fixation is superior.

Sorry the delay in getting back to you with a decision regarding your manuscript. I have now discussed everything with my colleagues and also with the referees.

So the issue remains that the quality of the PFA staining is not as good as it could be and that it is therefore difficult to conclude that the Glyoxal method is superior. For example take a look at the actin images shown in the Leyton-Puig et al. 2016 paper "PFA fixation enables artifact-free super-resolution imaging of the actin cytoskeleton and associated proteins." Biol Open, vol. 5 2016. The images in that paper look really good. So I agree with referees #2 and 4 regarding this point.

I can offer that should you be able to show that your protocol is superior to state-of-the-art PFA methods that I can offer to take a look at another version. The issue raised regarding the buffer (referee #2) also needs to be resolved. I would run the version by referee #2 again and would need the support from this referee. It is a bit unclear at this stage if you can resolve the concerns and so you should carefully consider your options.

Response to the referees:

Referee #1:

The majority of questions raised in the initial review have been answered by adding a large amount of new data. I nonetheless have two remaining points of concern that need to be tackled before publication:

Fig. S2 is not compelling in my view. What are the images on the top? It seems as if only a fraction of a cell is shown.

We are now showing the full images that we acquired.

Moreover, internalized transferrin is expected to exhibit a punctate vesicular pattern rather than the reticular pattern seen in the PFA fixed sample. How is this explained?

The experiment investigates cells that are exposed to transferrin during fixation. The reticular pattern is not seen in living cells, which implies that it must be an artifact due to fixation. We assume that it is due to the entry of free, unfixed transferrin into the cells, through the fixation-damaged plasma membrane, which is followed by their fixation onto different elements in the cell. It is probable that the transferrin molecules do not penetrate into organelles as efficiently as they diffuse into the cytosol, which results in a reticular pattern, with the spaces occupied by organelles remaining more free of transferrin.

Finally, I do not understand why the fixative remains present during the uptake as the question to address really is whether PFA fixed cells remain endocytosis competent longer than glyoxal fixed ones.

The experiment was performed to determine whether ligands such as transferrin can be taken up during fixation in glyoxal or PFA, either through specific endocytosis mechanisms, or through unspecific entry into the cells. The fixative is necessary for this experiment.

We have also tested whether the fixed cells are endocytosis-competent in the FM imaging experiments from Figure 1. Endocytosis was only apparent for the PFA-fixed cells.

The PIP2 stains in Fig. S8 do not live up to the standard of an EMBO J paper, irrespective of whether PFA or glyoxal is used as fixative. While publication of the Ms does not depend on this experiment I suggest that better and more compelling data using established lipid antibodies and their colocalization with PH probes are used to test preservation of lipid antigens.

We have used an established PIP2 antibody, which has been widely published. The results we obtained are fully consistent with the original stainings of such antibodies (Thomas et al., 1999).

We would like to point out that this is what such immunostainings appear like, once the cells are permeabilized with detergents, to enable antibody penetration. The permeabilization procedure extracts much of the lipids, and disturbs the PIP2 patterns.

Referee #2:

The revised version of the manuscript "Glyoxal is superior to PFA in immunostaining and nanoscopy" contains several additional experiments and clarifications, and is thus improved. Unfortunately, in my view, the bold claim that Glyoxal is a better fixative than PFA is not sufficiently substantiated by the data, also because for many experiments the effect is rather small or difficult to quantify and there seem to be some inconsistencies. Also, the effect is often overstated. Of course, it could be true that glyoxal is a better fixative than PFA, but if it is not, it would have negatively impact the many groups which might change their fixation protocols based on this manuscript.

To address this concern we decided to have our protocol tested, in an independent fashion, by multiple laboratories that are well-versed with immunostaining procedures. We have therefore asked 11 laboratories, from four countries, to compare glyoxal and PFA in immunostainings, using their favorite sample preparation and imaging procedures. The following laboratories were involved: Boyden (MIT Media Lab and McGovern Institute, Massachusetts, United States), Duncan (Heriot-Watt University, Edinburgh, UK), Hell (Max Planck Institute for Biophysical Chemistry, Göttingen, Germany), Lauterbach (Max Planck Institute for Brain Research, Frankfurt am Main, Germany), Lehnart (University Medical Center Göttingen, Germany), Moser (University Medical Center Göttingen, Germany), Outeiro (University Medical Center Göttingen, Germany), Rehling (University Medical Center Göttingen, Germany), Schwappach (University Medical Center Göttingen, Germany), Testa (KTH Royal Institute of Technology, Stockholm, Sweden), and Zapiec (Max Planck Research Unit for Neurogenetics, Frankfurt am Main, Germany). The results are now shown in the revised manuscript. In summary, all laboratories have found that glyoxal improves immunostainings, for a multitude of targets. A total of 56 different targets or conditions have been tested in the different laboratories, of which 31 were significantly better immunostained and/or preserved after glyoxal fixation. The immunostainings were comparable for 14 targets. A poorer immunostaining was only observed for 11 targets. A graphic overview is shown in the figure below.

We therefore conclude that glyoxal is indeed a better fixative than PFA, for the majority of the cellular targets investigated.

Major comments:

1. Effect of fixative vs effect of buffer: As the authors use different fixation buffers (addition of Ethanol, pH), it is not clear if the perceived differences are due to the fixative or the buffer. For example, PFA+EtOH clearly shows faster fixation than glyoxal, in contrast to their statement that glyoxal is a faster fixative than PFA.

PFA+EtOH does not fix faster than glyoxal. In Supplementary Figure 7, where the level of fixation is directly measured, we have stated that the addition of ethanol does not change the fixation speed of the PFA solution.

The referee perhaps refers to Supplementary Figure 1, where ethanol addition speeds up the penetration of the PFA solution into cells. This has little to do with the fixation speed, since the latter depends on the speed of the chemical reactions, and not just on the speed of penetration into cells.

Also, totally different pH values were used, which might be relevant, as also PFA might show an improved performance under low pH (eg. P 203 in: Puchtler, H., & Meloan, S. N. (1985). On the chemistry of formaldehyde fixation and its effects on immunohistochemical reactions. Histochemistry, 82(3), 201-204). Thus, as long as different buffers are used, I would not support the bold claim that PFA is better than glyoxal, rather the authors could call it an alternative fixation method. To make that claim, the authors have to substantiate that neither the pH, or the ethanol would make PFA as good or better than glyoxal. To address this point, the authors should repeat all experiments using the same buffer for PFA and glyoxal (also e.g. for Figure S7).

In the last version of our manuscript, we had demonstrated that:

- Ethanol does improve cellular penetration of PFA (Fig. S1)
- Ethanol does not improve cellular fixation (Fig. S7)
- Ethanol does not improve the morphology of fixed samples (Fig. S5)

To address the further comments of the reviewer, we have investigated further the effects of different pH values (4 and 5) on PFA fixation, along with an additional condition suggested by the editor (short PFA fixation, 10 minutes, 37°C, at pH 7). None of these conditions improved PFA fixation, as measured by SDS-PAGE experiments (New Fig. S7).

Furthermore, we have analyzed the effects of ethanol and different pH values on PFA fixation on immunostainings (New Fig. S1) and on the general cell morphology (Supp. Table 1). None of these conditions improved PFA fixation.

2. I not convinced that the authors use an optimal PFA protocol.

It is difficult to imagine that one PFA protocol can be optimal for all possible immunostainings.

Two opposite requirements need to be fulfilled by a fixation protocol:

- To fix as strongly as possible, to maintain the native morphology of the samples
- To allow optimal penetration of the antibodies into the samples

The stronger the fixation protocol, the less signal will be observed. This takes place, for example, when adding glutaraldehyde into the PFA solution. The shorter and milder the fixation (*e.g.* brief incubations with PFA), the more cellular proteins will be lost into the buffer during

permeabilization. This is optimal for actin fixation, for example, but results in poor fixation for membrane proteins.

We have tested here 5 protocols for PFA fixation (with and without ethanol, pH 7, 4 and 5), and our 11 collaborators have also used their various individual protocols. Glyoxal provided overall better results.

In Figure 2 they observe large-scale morphological changes even after 50 minutes, and have not even reached equilibrium (I would suggest plotting the curves until an equilibrium is reached). Even the glyoxal fixed cells still change after 20 minutes. Also, most of the examples in Figure S6 show the same correlation with PFA and glyoxal, but I would expect a large scale morphological change (Figure 2) to affect the positions of all organelles and thus markers, and therefore a much more pronounced difference in S6 for all markers.

Brightfield images of cells show the morphological changes in all organelles and membranes. As some elements remain mobile, such as flapping cell margins, which respond to fluid motion in the dish, the equilibrium will be only slowly reached.

The cells in figure S6 do show such changes: for example, the volume of the cytosol changes before and after fixation, the cells shrink and become flatter (see images below), which would be observed as a large-scale change in Figure 2. However, in figure S6 we have carefully re-focused and adjusted the images to show the same areas, and the same organelles, to minimize the effect, in order not to confuse the reader.

Figure. Comparison of pre- and post-fixation images of a cell expressing SNAP25-GFP. The images were acquired using an epifluorescence microscope, and were processed using a band pass filter, to sharpen the morphological features. While the images are overall similar, an overlay (left) indicates that little remains unchanged after fixation with PFA. The red frame in the overlay corresponds to the pre-fixation image; the green frame corresponds to the post-fixation image. Scale bar = 10 μ m

Additionally, in our own experiments when we fix on the microscope with PFA, we observe an immobilization of our structures of interest within 10-20 seconds.

This may depend on the structures that one images. Synapses still exocytose for many minutes in PFA (Smith and Reese, J Exp Biol 1980), and membrane proteins are extremely mobile even after 30 minutes of fixation (for example, Tanaka et al., Nature Methods, 2010), and in contrast to our own experience.

We have added here two movies from our own work, performed in 2008, in the laboratory of Stefan Hell (Max Planck Institute for Biophysical Chemistry, Göttingen, Germany), long before we had conceived the current project. They show the mobility of synaptic vesicles either in living neurons, or in PFA-fixed neurons, which had been exposed to PFA for 45 minutes. The vesicle movement is still high, indicating that many molecules and/or organelles are poorly bound to the rest of the cell, and remain mobile. The movies show 38 STED frames, taken at 28 frames-per-second. The frames are 1.8 μ m wide, and have been saved as animated GIF files (opened by, for example, Windows Internet Explorer), to keep the file size as small as possible.

The high mobility observed in PFA-fixed samples forced us to use 3% glutaraldehyde to fully fix our samples, for our negative (no mobility) controls (see Westphal et al., Science, 2008).

Minor comments

1. *Figure 5: I am not sure if objects/um² is a right measure, especially if it is not clear what is an object, and if more objects (better preservation) or fewer objects (lower fragmentation) is better. From the images, it is not clear which condition is better, as again we do not know the ground truth. We have explained the nature of the objects, for the different target proteins, in the text. However, to avoid further discussion on this subject, we have removed the particular analysis to the supplement, and we left in the main text a simple analysis of signal intensity, compared between PFA- and glyoxal-fixed samples. This shows that glyoxal should be preferred to PFA, for most stainings (figure 4).*

2. *Figure 6/7: Averaging for superresolution microscopy might become a standard for well-defined complexes (as the NPC), but for highly heterogeneous structures, which are not regular at all, I still find it naïve, and the structures in Figure 6 are like that. Even on regular structures the averaging algorithms are not very robust and tend to overlay bright spots. Also, the averages in Figure 6 often produce very different structures than visible in the single images (Fig 4, 5). Thus, I don't think this analysis helps to compare the fixation protocols.*
Please see the answer to minor comment #1.

3. *Figure S1: how can PFA with EtOH be so much faster than glyoxal, if PFA alone does not permeabilize the membrane at all? if this data is true, it would indicate that glyoxal slows down membrane permeabilization by EtOH - is that possible?*
There is no significant difference, as indicated by the error bars.

4. *Figure S2: How was the membrane segmented for the PFA?*
We performed lines scans across the border of the cell, and measured the signal in the region at the edge of the cell, compared to that found deeper within the cell.
Maybe transferrin does not bind any more to the PFA-fixed receptors, and just enters the cell?
This is highly probable. This does not happen in glyoxal, where PFA is fixed onto the plasma membrane, before it can enter the cell, again indicating that PFA is a poorer fixative.
Were both images taken at a focal plane through the centre of the cell?
Yes.

5. *Figure S3: this is expected if the membranes are permeabilized by EtOH, but does characterize PFA vs glyoxal.*
We agree with the comment of the reviewer, but the experiment was requested by Reviewer 1, and we therefore needed to perform it.

6. *Figure S4: seems to contradict Figure 2, as clearly transferrin uptake is stopped within 1 minute.*
This is not the case. There is no transferrin in solution in Figure S4, so one cannot tell if the uptake is stopped. Endocytosis still goes on in PFA-fixed cells, as shown in Fig. 1.

7. *Figure S5: why not use the correlation-based analysis as before? The increase in brightness upon glyoxal fixation is unexpected.*
The increase in brightness was not significant. It was probably due to re-focusing between imaging sessions.
This figure was made in the initial (pre-revision) version of the manuscript, before we introduced correlation-based analysis. A correlation-based analysis of the same target (mitochondria) was shown in the original Supplementary Fig. 6.

8. *Figure S6: I would suggest showing the image for Synaptophysin, as this is the only protein where a difference is observed. Also, please choose similar cells for PFA and glyoxal.*
We now show example images for Synaptophysin.

9. *Figure S8: As to my knowledge PIP2 does not have an amine group, probably the authors test lipid immobilization by protein fixation, not fixation of the lipids themselves.*
This is certainly true. We did not claim otherwise. We probably also test epitope availability: not just whether the PIP2 molecules are immobilized, but also whether they are available for antibody binding.

10. Figure S9: how exactly was the intensity quantified? In the image, the glyoxal shows a diffuse bright patch. The authors should make sure that their quantification is not influenced by unspecific labelling.

We calculated the total intensity per cellular ROI. There was no staining in the untransfected cells (see new Figure S9).

11. Figure S12: please show the profile for HPF.

We have measured it, and it is now shown in the respective figure. It is virtually identical to those from the glyoxal experiments, and different from the PFA experiment.

12. Table S1: Please include the PFA only.

We have greatly enlarged Table S1, and added different PFA conditions.

Referee #4:

The authors claim to have discovered a new fixative (Glyoxal) that is less toxic than PFA and is a better preservative of cellular morphology. Unfortunately, the data shown for PFA fixation is below standard, suggesting that the authors have not optimized their fixation and/or imaging conditions in cells that are PFA fixed. Without this comparison, it is impossible to know whether or not glyoxal is superior. None of the images in this paper are of high quality. The authors must first show an example of optimally PFA fixed and stained images before I can believe that glyoxal fixation is superior.

As replied also to the second reviewer, it is difficult to imagine that one PFA protocol can be optimal for all possible immunostainings. Two opposite requirements need to be fulfilled by a fixation protocol:

- To fix as strongly as possible, to maintain the native morphology of the samples
- To allow optimal penetration of the antibodies into the samples

The stronger the fixation protocol, the less signal will be observed. This takes place, for example, when adding glutaraldehyde into the PFA solution. The shorter and milder the fixation (*e.g.* brief incubations with PFA), the more cellular proteins will be lost into the buffer during permeabilization. This is optimal for actin fixation, for example, but results in poor fixation for membrane proteins.

We have tested here 5 protocols for PFA fixation, and did not manage to find one that was consistently better than glyoxal.

To address this concern further we decided to have our protocol tested, in an independent fashion, by multiple laboratories that are well-versed with immunostaining procedures. We have therefore asked 11 laboratories, from four countries, to compare glyoxal and PFA in immunostainings, using their favorite sample preparation and imaging procedures. The results are now shown in the revised manuscript. In summary, all laboratories have found that glyoxal improves immunostainings, for a multitude of targets (please see pages 2 and 3 of this reply).

3rd Editorial Decision

12 September 2017

Thanks for submitting your revised manuscript to The EMBO Journal. Your study has now been re-reviewed by referee #2 and I am happy to say that all is fine. I am therefore very pleased to accept the manuscript for publication in The EMBO Journal.

REFEREE REPORT

Referee #2:

As the authors tuned down their claims and showed data from 11 other labs using Glyoxal, this manuscript might be as good as it can get and could be considered for publication.

Corresponding Author Name: Silvio O. Rizzoli

Manuscript Number: EMBOJ-2016-95709